# PIXEL MOTION AS UNIVERSAL REPRESENTATION FOR ROBOT CONTROL

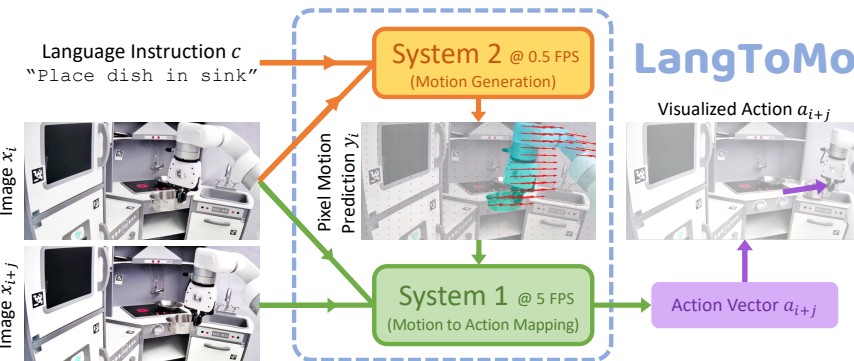

Figure 1: Dual-System VLA Framework, LangToMo, with pixel motion representations.

## ABSTRACT

We present LangToMo, a vision-language-action framework structured as a dual-system architecture that uses pixel motion forecasts as intermediate representations. Our high-level *System 2*, an image diffusion model, generates text-conditioned pixel motion sequences from a single frame and past motion to guide robot control. Pixel motion—a universal, interpretable, and motion-centric representation—can be extracted from videos in a weakly-supervised manner, enabling diffusion model training on any video-caption data. Treating the generated pixel motion as largely embodiment-agnostic *universal representations*, our embodiment-aware *System 1* module translates these into robot actions via motion-to-action mapping functions, which can be either hand-crafted or learned with minimal supervision. System 2 operates as a high-level policy applied at sparse temporal intervals, while System 1 acts as a low-level policy at dense temporal intervals. This hierarchical decoupling enables flexible, scalable, and generalizable robot control under both unsupervised and supervised settings, bridging the gap between language, motion, and action. Visualizations at `anonymous.4open.science/w/LangToMo`.

## 1 INTRODUCTION

Translating open-ended natural language instructions into robot actions is a cornerstone of flexible robot control. We identify two key requirements to enable this: (i) universal embodiment-agnostic representations that support operating diverse embodiments (Nair et al., 2022; Ren et al., 2025a; Zheng et al., 2025), and (ii) benefiting from video-language data without action labels (Du et al., 2023b; Gu et al., 2023; Black et al., 2023; Ko et al., 2023; Cheang et al., 2025; Lee et al., 2025). We explore their intersection, proposing LangToMo, a vision–language–action (VLA) framework structured as a *dual-system architecture*, inspired by dual-process theories of cognition (Kahneman, 2011) and recent hierarchical robotics frameworks (Belkhale et al., 2024; Black et al., 2024; Shi et al., 2025b; Nvidia et al., 2025; Intelligence et al., 2025). We use a high level *System 2* module, implemented as an image diffusion model conditioned on a visual observation and textual instruction, to generate pixel motion: our intermediate robot action representation that is universal and interpretable. Subsequently, our embodiment-aware low level *System 1* implemented as a vision transformer or hand-crafted algorithm deterministically projects these action representations into executable robot actions.

We adopt pixel motion—displacement of pixels in a frame—as our *universal motion representation*, because it is more agnostic (e.g. than RGB; see Appendix L) to embodiments, viewpoints, and tasks. By predicting pixel motion instead of full RGB images, LangToMo captures essential motion patterns more efficiently (i.e. with less training data, see Section 4.1) than text-to-video generation (Du et al., 2023b; Ko et al., 2023; Gu et al., 2023; Black et al., 2023). In contrast to operating with sparse point tracks (Yuan et al., 2024a; Wen et al., 2023; Xu et al., 2024; Bharadhwaj et al., 2024b), our dense pixel motion representation can capture both manipulator and object movements (see Figure 2). Our pixel motion features also retain the inherent 2D structure of the visual domain unlike prior work modeling pixel trajectories as 1D point tracks (Wen et al., 2023; Xu et al., 2024). Moreover, dense pixel motion can be freely computed from videos using off-the-shelf algorithms like RAFT (Teed & Deng, 2020), enabling scalable, weakly supervised training on large video-caption datasets, similar to prior work on predictive world models (Gu et al., 2023; Black et al., 2023; Zhang et al., 2025).

Optical flow, a measure of pixel motion (PM) between frame pairs, has been leveraged to enhance realistic motion in video generation (Liang et al., 2024a; Koroglu et al., 2024; Chefer et al., 2025), including in the robotics domain (Gao et al., 2025). PM calculated from current and future frames is used for robot control in Ko et al. (2023); Bharadhwaj et al. (2024a), further establishing the promise of this direction. In contrast, we directly generate PM from language and a single current frame (without access to future frames) using our System-2 module, offering greater data efficiency and performance (see Tables 2 to 4). Our predicted PM serves as an interpretable intermediate representation for downstream systems (e.g., our System-1), enabling even unsupervised control via hand-crafted mappings. Alternate motion signals in image-space are used in works like Sudhakar et al. (2024); Shridhar et al. (2024); Huang et al. (2024); Shi et al. (2025a), but they rely on explicit dense annotations limiting training scalability, unlike our System-2 formulation. Generating PM from a single image has also been explored (Walker et al., 2015; Gao et al., 2018; Aleotti et al., 2021), but with no language conditioning. In contrast, our System-2 module generates PM conditioned on both visual and textual cues with no access to future frames.

Sequences of PM generated by our System 2 are then transformed into robot actions via *System 1*, a fast and deterministic controller. Specifically, System 1 consists of motion to action mappings that are *embodiment aware*. We explore two instantiations of System 1: (a) learning mappings directly from limited expert demonstrations, and (b) hand-crafting mappings by leveraging the interpretable nature of pixel motion (motivated by Ko et al. (2023)). Connecting System 1 and System 2 forms our overall language-conditioned robot control framework, LangToMo. This hierarchical formulation allows operating the expensive high-level System 2 at sparse temporal intervals while invoking the lightweight low-level System 1 at dense temporal intervals for efficient inference. This also allows independent training of each system, leading to better overall training efficiency.

In summary, our contributions are as follows:

- **Universal Action Representation:** 2D structured, language-conditioned, dense pixel motion as a learnable, interpretable, and manipulator-motion focused representation for robot control.
- **Simple & Scalable Learning:** mapping natural language actions to motion representations (pixel motion sequences) with a history-aware conditional diffusion model trained on any video-caption data, without requiring pixel-level or action trajectory annotations.
- **Robotics Application:** conversion of learned action representations into action policies with minimal supervision, enabling operation under zero-shot and even unsupervised settings.

We evaluate LangToMo on both simulated and real-world environments, highlighting its effectiveness and generality across diverse robot control tasks.

## 2 RELATED WORK

**Learning from Videos:** Robot learning has a rich history of leveraging videos to extract sub-goal information, learn strong representations, or build dynamics models for planning (Lee & Ryoo, 2017; Finn & Levine, 2017; Sun et al., 2018; Kurutach et al., 2018; Pari et al., 2022; Nair et al., 2022; Shao et al., 2021; Chen et al., 2021; Bahl et al., 2022; Sharma et al., 2019; Du et al., 2023b; Sivakumar et al., 2022; Sudhakar et al., 2024; Ko et al., 2023; Hu et al., 2025; Ren et al., 2025a). Methods like LAPA, IGOR, Villa-X, UniVLA (Ye et al., 2025; Chen et al., 2024; Bu et al., 2025; Chen et al., 2025b) also allow pretraining from actionless videos, but their latent representations require

Table 1: **Unique Features of LangToMo.** *Dual System*: Decoupled architecture with System 1 & 2 modules trained separately and inference at distinct frequencies. *Dense Motion*: Uses no heuristic / training based point sub-sampling. *2D Structure*: Pixel motion as 2D grid instead of coordinate based 1D point sequence. Prior work A to G are Yuan et al. (2024a), Gao et al. (2025), Wen et al. (2023), Xu et al. (2024), Bharadhwaj et al. (2024b), Bharadhwaj et al. (2024a), Hu et al. (2025) respectively.

| Feature | Ours | A | B | C | D | E | F | G |
|---|---|---|---|---|---|---|---|---|
| Past Aware | ✓ | ✗ | ✓ | ✗ | ✗ | ✗ | ✗ | ✗ |
| Dual System | ✓ | ✗ | ✗ | ✓ | ✓ | ✓ | ✗ | ✓ |
| Dense Motion | ✓ | ✗ | ✓ | ✗ | ✗ | ✗ | ✗ | ✗ |
| 2D Structure | ✓ | ✗ | ✓ | ✗ | ✗ | ✗ | ✗ | ✓ |
| Text Condition | ✓ | ✓ | ✓ | ✓ | ✓ | ✗ | ✓ | ✓ |

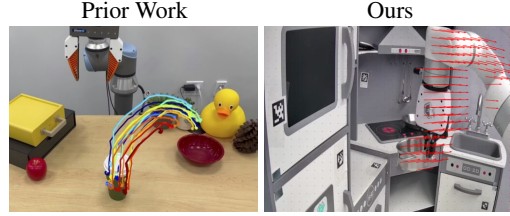

Prior Work     Ours

Figure 2: **Dense Motions:** Most prior work that use pixel trajectories focus on a subset of pixels often limited to objects of interest. The example from Xu et al. (2024) (left) focuses on the cup movement, but ignores important action information relevant to manipulator movement. In contrast, our proposed LangToMo generates motion of *every pixel* in an image, accounting for both object and manipulator movements (right).

fine-tuning on robot action trajectories. Another line of works like HiRT, LCB, and OpenHelix (Zhang et al., 2024; Shentu et al., 2024; Cui et al., 2025) leverage dual system architectures with LLM backbones and use intermediate latent features as connection between the two systems. Instead, we explore image diffusion backbones as our larger system given their suitability for spatial grounding (i.e. image diffusion architectures are designed to generate images). Further more, in contrast to these, methods like AVDC and ours use interpretable optical flow representations which allows action-free robot control even for the downstream tasks (Ko et al., 2023). Additionally, our method also benefits from human demos (much cheaper to collect than robot demos Cheang et al. (2025)) in downstream real world tasks. Several recent works learn representations connected to language modality from video-caption data (Du et al., 2023b; Sudhakar et al., 2024; Ko et al., 2023; Hu et al., 2025), but depend on additional action-trajectory annotations, pretrained segmentation models, or task-specific heuristics for robot control. We explore a similar direction, learning language-conditioned motion representations from video-caption data. In contrast to these works, our LangToMo learns representations that are *interpretable* and *motion-focused*, which we use for robot control with no additional supervision. Our focus on pixel motion also allows learning more generalizable representations with less data.

**Pixel Motion to Actions:** Robot navigation and control, especially in the context of aerial drones, has long benefited from optical flow representations (de Croon et al., 2021; Lee et al., 2020; Hu et al., 2024; Argus et al., 2020a), inspired by animal perception systems that use optical flow for stable control and movement (Götz, 1968; Arnold, 1974; Ros & Biewener, 2016; Baird et al., 2021). Video self-supervised learning has also extensively leveraged optical flow to learn motion representations (Han et al., 2020; Sharma et al., 2022). In robot control, trajectories of pixel subsets (Yuan et al., 2024a; Wen et al., 2023; Xu et al., 2024; Bharadhwaj et al., 2024b) have been used as intermediate representations, but often limit focus to specific image regions or objects, ignoring global information such as manipulator movement (e.g. see Figure 2). In contrast to prior work, our LangToMo models dense pixel motion (focusing on both object and manipulator movement) conditioned on textual action descriptions and current visual observations with no future frame dependency (see Table 1).

**Diffusion-Based Motion Generation:** Diffusion models have emerged as powerful generative frameworks capable of capturing complex data distributions through iterative denoising processes (Ho et al., 2020; 2022; Ramesh et al., 2022; Zhang et al., 2023; Singer et al., 2022; Villegas et al., 2022; Ge et al., 2022; Kumari et al., 2023; Zhang et al., 2022; Ren et al., 2022; Li et al., 2024c; Chen et al., 2023; Janner et al., 2022; Du et al., 2023a; Liu et al., 2023; Wang et al., 2023b; Chi et al., 2023; Shridhar et al., 2024; Chefer et al., 2025). Some works directly predict optical flow from image pairs (Saxena et al., 2023; Luo et al., 2024), which are well-defined inputs. In contrast, LangToMo generates pixel motion from a single image and language command, capturing the multi-modal nature of future motions. By also conditioning on past motion (extracted from current observations), our approach introduces temporal grounding, making it well-suited for robot control.

**Language-Conditioned Robotic Manipulation:** Several recent works use vision-language models for robot control (Brohan et al., 2023b;a; Padalkar et al., 2023; Reed et al., 2022; Wu et al., 2023; Octo Model Team et al., 2024; Driess et al., 2023; Kim et al., 2024; Yuan et al., 2024b; Niu et al., 2024; Zheng et al., 2024; Li et al., 2024d; Zawalski et al., 2024; Hu et al., 2025; Sudhakar et al., 2024; Ko et al., 2023; Tian et al., 2024; Jeong et al., 2025) taking advantage of large-scale training

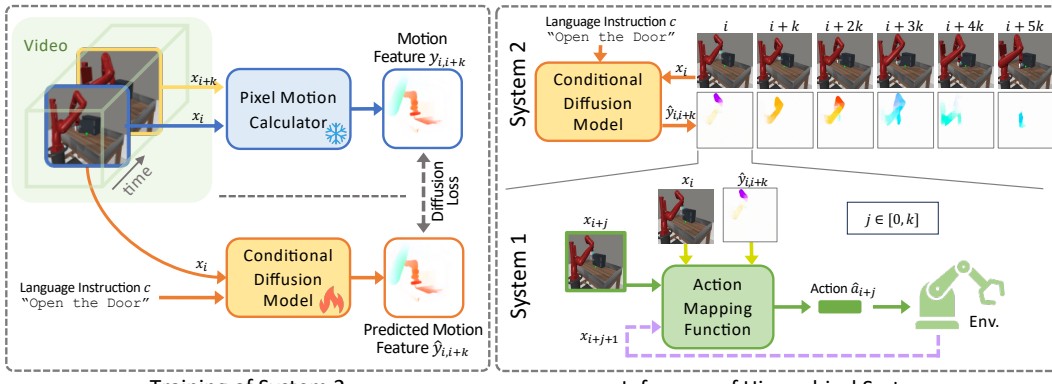

Figure 3: **Overview of LangToMo:** (Left) We learn to forecast pixel motion as universal motion features from video-caption pairs using scalable, self-supervised training of a diffusion model. (Right) Our *System 2* forecasts motion at sparse intervals $(k)$, while *System 1* maps it to dense action vectors at $j$ intervals $(j < k)$.

with web-scale vision-language data. In contrast to prior work using sequential language models, we learn motion representations under weak supervision (only video-caption data) using zero action trajectory annotations. We also utilize an image diffusion model similar to Hu et al. (2025); Sudhakar et al. (2024); Ko et al. (2023) but differ by learning universal and interpretable motion representations directly, which even allows conversion to robot actions directly with no further training.

## 3 METHODOLOGY

We tackle the problem of robot control from natural language instructions by introducing a two-stage framework. Language and visual inputs are first encoded into pixel motion based representations, which are then decoded into robot actions. This dual-system architecture comprises: *System 2*, a conditional image diffusion model that generates embodiment agnostic motion features at sparse temporal intervals acting as a high-level controller; and *System 1*, an embodiment aware low-level controller that maps these pixel motions to executable robot action vectors. An overview of our framework, LangToMo, is shown in Figure 3.

**Pixel Motion:** Our use of the term "*Pixel Motion*" refers to future optical flow between the current frame and a possible future frame. Optical flow is between two frames that are fixed, but in our case the future frame can be one of many possibilities - so we use the term Pixel Motion (PM) instead.

### 3.1 SYSTEM 2: PIXEL MOTION FORECAST

Optical flow estimation from frame pairs is a well-defined problem (exact solutions exist) that has been extensively studied (Liu et al., 2019; Teed & Deng, 2020; Xu et al., 2022; Luo et al., 2024). In contrast, estimating pixel motion–the trajectory of pixels towards a goal state–from a single image and language instruction is inherently multi-modal: a caption-frame pair may correspond to multiple valid pixel motions, each representing a different trajectory toward the goal. We use this challenging task as our training objective: learning a mapping from *language to motion*. Furthermore, we incorporate temporal context by conditioning on the motion of a previous state.

Consider a video clip $\boldsymbol{x} \in \mathbb{R}^{t \times h \times w \times c}$ with $t, h, w, c$ for frames, height, width, and channels respectively. Also consider an embedding vector, $\boldsymbol{c}$ representing the paired caption for that clip. Denoting the $i$-th frame of video as $\boldsymbol{x}_i$, we define pixel motion, $\boldsymbol{y}_{i,i+k}$, that corresponds to motion between frames $\boldsymbol{x}_i \rightarrow \boldsymbol{x}_{i+k}$ where $k$ is a constant. Our language to motion mapping function, $\mathcal{D}$ becomes,

$$\hat{\boldsymbol{y}}_{i,i+k} = \mathcal{D}\left(\boldsymbol{x}_i, \boldsymbol{y}_{i-k,i}, \boldsymbol{c} \mid \theta\right) \tag{1}$$

where $\hat{\boldsymbol{y}}_{i,i+k}$ is the predicted motion representation from the $i$-th state to $(i + k)$-th state *without* knowing frame $\boldsymbol{x}_{i+k}$, and $\theta$ are learnable parameters.

We reiterate the multi-modal output aspect of our mapping described in Equation (1) (i.e. one to many mapping due to multiple optimal $\hat{\boldsymbol{y}}_{i,i+k}$). Diffusion models have shown excellent abilities to model

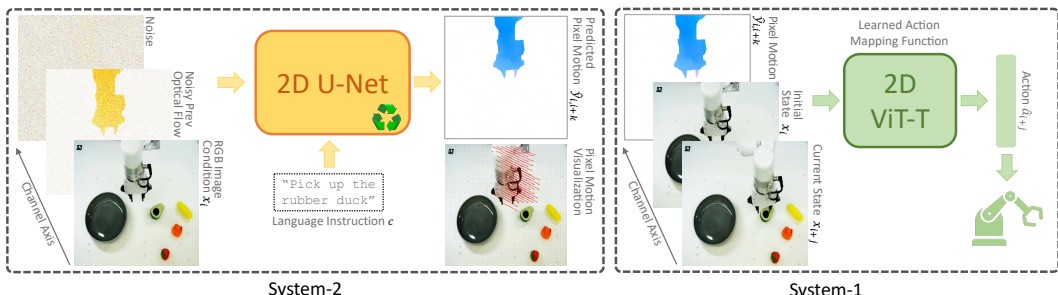

Figure 4: **LangToMo Architecture:** (Left) Diffusion model generates pixel motion conditioned on RGB image, prior motion, and caption. Visualized predictions are overlaid as arrows. (Right) ViT-T network maps predicted motion to robot actions in supervised setting, conditioned on initial/current states and target motion.

such distributions (Dhariwal & Nichol, 2021; Chi et al., 2023). Considering the 2D structure present in our images and pixel motion, for $\mathcal{D}$ we elect to utilize a 2D conditional U-Net based diffusion model (Ramesh et al., 2022) operating at pixel level. Our goal is to learn a set of parameters, $\theta$ for this diffusion model based mapping as,

$$\arg\min_{\theta} ||\boldsymbol{y}_{i,i+k} - \mathcal{D}\left(\boldsymbol{x}_i, \boldsymbol{y}_{i-k,i}, \boldsymbol{c} \mid \theta\right)||_2 \qquad (2)$$

that allows our language to motion mapping to perform instruction based robot control. Next we dive into the learning process of our diffusion based implementation for this mapping function.

## 3.2 DIFFUSION BASED MOTION REPRESENTATION LEARNING

**Background:** Diffusion Models generate data by progressively denoising corrupted signals, optionally conditioned on a goal input. While inference follows this iterative refinement process, training is conducted more efficiently using parallel denoising steps: the model is trained to predict less noisy versions of intermediate corrupted signals generated from clean data, a procedure analogous to teacher forcing (more details in Appendix E).

**Architecture:** The defacto architecture for diffusion based conditional image generation is the 2D conditional U-Net (Ronneberger et al., 2015), which maps between 2D RGB images with an embedding based conditioning through cross-attention in the model intermediate layers. Basing off this setup, we modify the input and output heads to process 7 and 2 channel tensors respectively (instead of default 3 channel RGB). Two of the input channels and the two output channels correspond to our pixel motion target (noise input and clean output). The remaining 5 input channels correspond to our 2D-structured conditions: previous pixel motion (2 channels) and current state image (3 channels). These conditional inputs are not subject to the standard noise corruption schedule during training or inference (details in Appendix E). The textual embedding is provided as the default embedding condition. Our channel modification to accommodate additional structured conditions allows minimal re-design, retaining the general structure of the U-Net that is known to excel at 2D generative modeling. Such input channel concatenation based conditioning has been used in diffusion literature for different tasks (Saxena et al., 2023; Ho et al., 2022) and is inspiration for our design. We illustrate this architecture in Figure 4 (Left).

**Calculating Pixel Motion Ground-truth:** We utilize the RAFT algorithm (Teed & Deng, 2020) to calculate our target pixel motion $\boldsymbol{y}_{i,i+k}$, using frames $\boldsymbol{x}_i$ and $\boldsymbol{x}_{i+k}$. This is an efficient iterative algorithm that calculates a good estimate of optical flow, in other words, pixel motion. Each pixel motion, $\boldsymbol{y}_{i,i+k} \in \mathbb{R}^{h \times w \times 2}$, contains two channels for spatial directions, that are normalized to a $(0, 1)$ range. All motion is represented within this 2D space - extensions to a third depth dimension are left as a future direction. Our experiments indicate the sufficiency of such 2D spaces to encode motions relevant to robot actions. We note that given the presence of background motions in both natural and simulation images (e.g. shadows moving with objects), this target pixel motion contains noise that is not directly relevant to the underlying motion, underscoring the challenging nature of our self-supervision objective.

**Previous Pixel Motion Representation:** Besides the current observation and 2-channel noise, the other input signal to our mapping function is past pixel motion. Motivated by success of teacher

forcing both language (Radford et al., 2019) and video (Song et al., 2025b) generation, we use the target pixel motion of previous time steps during our System-2 training. We also note the importance of representing pixel motion relative to current state as our mapping function is conditioned on the current image (details in Appendix C). Similar findings are observed in image-pair based optical flow calculation literature (Ko et al., 2023).

**Language Instruction Embeddding:** The cross-attention based conditioning of our mapping function is the natural language based action description that is used to control the generated motions. Following prior robotics literature (Padalkar et al., 2023), we use a Universal Sentence Encoder model (Cer et al., 2018) to convert textual instructions to fixed size embedding vectors. This embedding model is trained to capture sentence level meanings. We use an off-the-shelf pretrained version, keeping all model parameters unchanged (more details in Appendix D).

**Training:** Our training uses the standard diffusion denoising objective (Ho et al., 2020) between predicted ($\hat{y}_{i,i+k}$) and target ($y_{i,i+k}$) pixel motion. The conditional 2D inputs, $x_i$ and $y_{i-k,i}$ are not subject to a noising schedule. The image condition, $x_i$, remains uncorrupted while the previous pixel motion, $y_{i-1,i}$, is set to random noise or a partially corrupted version to align with inference settings. We also introduce zero motion to ends of videos such that when textual instruction is complete, those visual states map to zero motion. More details in Appendix E.

**Inference:** We forecast pixel motion from $i$ to $i + k$ timestamp using a 25-step DDIM schedule with only the current image observation $x_i$. At the initial step, the model only takes the image $x_i$ (state observation), language instruction $c$, and zero vector as the previous pixel motion. For subsequent steps, the previous motion condition is calculated using RAFT with $x_{i-k}$ and $x_i$ frames, enabling sequential pixel motion generation that drives the system toward fulfilling the language command.

### 3.3 System 1: Pixel Motion to Action Mapping

Our System 2 produces pixel motion conditioned on a given state-instruction pair. We next detail how these pixel motion representations are mapped into action vectors that directly control the robot. Consider a mapping function, $\mathcal{F}$, operating at dense temporal intervals:

$$\hat{a}_{i+j} = \mathcal{F}\left(\hat{y}_{i,i+k}, x_i, x_{i+j}\right), \tag{3}$$

where $j \in [0, k]$, $i$ is a multiple of $k$ (for a hyperparameter $k$), and $\hat{a}_{i+j}$ denotes the predicted action vector for the $(i + j)$-th state. An overview of this formulation is shown in Figure 3 (right).

While *System 2* is trained as a general-purpose motion generator across diverse embodiments, viewpoints, and environments, action vectors $a_i$ are inherently embodiment-specific. Hence, we design *embodiment-aware* mapping functions to serve as *System 1 (Action Mapping)*, that are capable of converting pixel motion into executable robot actions.

**Learned Mapping:** We implement a neural network-based mapping function that can be trained using ground-truth action trajectories. Given the 2D spatial structure of the inputs to $\mathcal{F}$ (i.e., $\hat{y}_{i,i+j}$, $x_i$, $x_{i+j}$), we channel-concatenate them and feed the resulting tensor to a lightweight vision transformer to predict action vectors. This architecture is illustrated in Figure 4 (right). The network is trained on a limited amount of embodiment-specific demonstration data. Connecting this learned *System 1* with *System 2* following Equation (3), we obtain a complete pipeline for language-conditioned robot control. We refer to the resulting system, which uses a supervised learned mapping, as LTM-S.

**Hand-Crafted Mapping:** The interpretable nature (see Appendix F) of pixel motion also enables hand-crafted designs for $\mathcal{F}$. We refer to the resulting pipeline based on hand-crafted mappings as LTM-H. For simulated environments where ground-truth segmentations and depth maps are available, we follow the methodology in (Ko et al., 2023) to define action mappings, ensuring a fair evaluation of the utility of our pixel motion predictions compared to prior works. For real-world robot control, we construct viewpoint-specific hand-crafted mappings following (Li et al., 2024d). Further details on both learned and hand-crafted mappings are provided in Appendix F.

We highlight how our System 1 operates at a frequency different to our System 2, allowing a balance between efficiency and dense control. Our System 1 is also designed to be lightweight, given how it performs an almost deterministic mapping.

Table 2: **Zero-Shot Transfer on Real World Tasks:** We directly deploy our pretrained model (with no fine-tuning) on real world tasks. We highlight our strong performance compared to baselines in this highly challenging setting. Evaluations follow Li et al. (2024d).

| Method | Video Only Training | T1 | T2 | T3 | T4 | Avg |
|---|---|---|---|---|---|---|
| RT-2 Style | ✗ | 0 | 0 | 0 | 0 | 0 |
| LLaRA | ✗ | 40 | 20 | 10 | 20 | 22.5 |
| AVDC | ✓ | 0 | 0 | 0 | 0 | 0 |
| GPT-4o | ✓ | 20 | 30 | 10 | 15 | 18.8 |
| LTM-H (ours) | ✓ | 40 | 30 | 35 | 30 | 33.8 |

Table 3: **Finetuned on Real World:** LangToMo benefits from both robot (RD) and human (HD) demonstrations highlighting the embodiment agnostic nature of our Sys-2, in contrast to prior work.

| Method | Data | T1 | T2 | T3 | T4 | Average |
|---|---|---|---|---|---|---|
| RT-2 Style | - | 0 | 0 | 0 | 0 | 0 |
| LLaRA | - | 70 | 80 | 55 | 55 | 65.0 |
| AVDC | RD | 10 | 20 | 0 | 0 | 7.5 |
| AVDC | RD+HD | 0 | 0 | 0 | 0 | 0.0 |
| LTM-H (ours) | HD | 40 | 35 | 40 | 30 | 36.3 |
| LTM-H (ours) | RD | 80 | 70 | 65 | 60 | 68.8 |
| LTM-H (ours) | RD+HD | 80 | 75 | 65 | 65 | 71.3 |

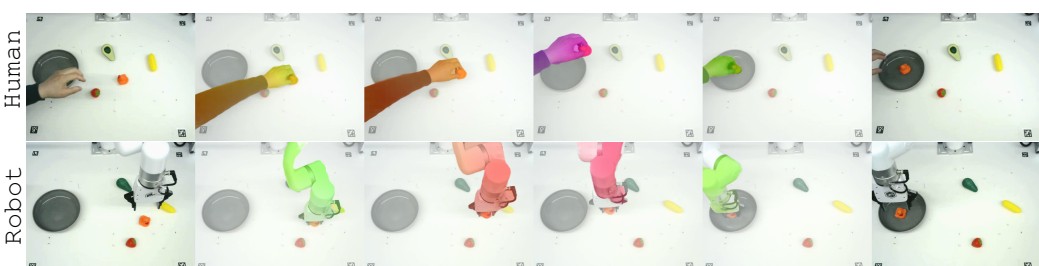

Figure 5: **Human (HD) & Robot (RD) Demonstrations:** We visualize frames from two sample demonstrations on our real world environment. Pixel motion (visualized in HSV color space: direction as hue, magnitude as intensity) overlaid on intermediate frames. These human (top) and robot (bottom) demonstrations can both be used to fine-tune our System-2, highlighting a unique aspect of LangToMo. Both examples use the same caption `"Pick up the rubber duck and place on the bowl."`

## 4 EXPERIMENTAL RESULTS

We conduct experiments on 15 task styles spanning both simulated and real-world environments to highlight the strong performance of our proposed LangToMo framework. We also present multiple ablations to justify key design choices within our method.

**Implementation Details:** Our framework consists of *System 2 (Motion Generation)* containing a diffusion model, and *System 1 (Action Mapping)* containing either a learned or hand-crafted mapping function. We pretrain the diffusion model on a subset of the OpenX dataset (Padalkar et al., 2023), followed by optional fine-tuning on downstream task datasets. Pretraining is performed for 300,000 iterations with a learning rate of 1e-4, following a cosine learning rate schedule with 500 warmup steps, using 8 A100 GPUs (48GB) with a per-device batch size of 32 samples. Fine-tuning is performed for 100,000 iterations on 4 A5000 GPUs (24GB) with a batch size of 32 and a learning rate of 1e-5, again following a cosine schedule with 500 warmup steps. The learned action mapping (System 1) is trained separately using a vision transformer for 10,000 iterations on a single A5000 GPU with a batch size of 128 and a learning rate of 1e-4. During inference of our System 2 diffusion model, we use a DDIM scheduler with 25 steps to generate flow sequences, starting from noise. For each invocation of System 2, we run System 1 for 10 control steps (or until convergence in the hand-crafted setting). Further details of our hierarchical inference pipeline in Appendix H.

### 4.1 REAL-WORLD ENVIRONMENT

We first evaluate on four styles of real world tasks using the xArm Table Top environment constructed following Li et al. (2024d). We select this environment and task styles for its ease of fair comparison to prior work, interpretable action dynamics (each state suggests a clear next motion), and demands for visual grounding, semantic understanding, and distractor robustness. The tasks involve object manipulations specified by language commands (details in Appendix G).

**Training & Evaluation:** We train *System 2* on the OpenX subset, followed by optional fine-tuning on demonstrations from the real-world environment. We collect 10 tele-operated robot demonstrations (RD) and 50 human demonstrations (HD) per task style. We replicate AVDC (Ko et al., 2023) by

Table 4: **Results on MetaWorld Environment:** We report the mean success rate across tasks. Each entry of the table shows the average success rate aggregated from 3 camera poses with 25 seeds for each camera pose.

| | door-open | door-close | basketball | shelf-place | btn-press | btn-top | faucet-close | faucet-open | handle-press | hammer | assembly | Overall |
|---|---|---|---|---|---|---|---|---|---|---|---|---|
| BC-Scratch | 21.3 | 36.0 | 0.0 | 0.0 | 34.7 | 12.0 | 18.7 | 17.3 | 37.3 | 0.0 | 1.3 | 16.2 |
| BC-R3M | 1.3 | 58.7 | 0.0 | 0.0 | 36.0 | 4.0 | 18.7 | 22.7 | 28.0 | 0.0 | 0.0 | 15.4 |
| Diffusion Policy | 45.3 | 45.3 | 8.0 | 0.0 | 40.0 | 18.7 | 22.7 | 58.7 | 21.3 | 4.0 | 1.3 | 24.1 |
| UniPi (With Replan) | 0.0 | 36.0 | 0.0 | 0.0 | 6.7 | 0.0 | 4.0 | 9.3 | 13.3 | 4.0 | 0.0 | 6.1 |
| Im2Flow2Act | 0.0 | 0.0 | 0.0 | 4.0 | 6.3 | 0.0 | 7.3 | 4.7 | 0.0 | 0.0 | 0.0 | 2.0 |
| ATM | 75.3 | 90.7 | 24.0 | 16.3 | 77.3 | 76.7 | 50.0 | 62.7 | 92.3 | 4.3 | 2.0 | 52.0 |
| $\pi_0$ | 0.0 | 0.0 | 0.0 | 5.0 | 8.4 | 0.0 | 6.8 | 9.2 | 0.0 | 0.0 | 0.0 | 2.7 |
| AVDC (Flow) | 0.0 | 0.0 | 0.0 | 0.0 | 1.3 | 40.0 | 42.7 | 0.0 | 66.7 | 0.0 | 0.0 | 13.7 |
| AVDC (Default) | 72.0 | 89.3 | 37.3 | 18.7 | 60.0 | 24.0 | 53.3 | 24.0 | 81.3 | 8.0 | 6.7 | 43.1 |
| AVDC (PT) | 72.0 | 88.7 | 37.3 | 18.7 | 58.7 | 24.3 | 53.3 | 24.0 | 81.3 | 8.0 | 6.7 | 42.9 |
| LTM-H (Ours) | 76.0 | 94.7 | 38.0 | 15.3 | 82.0 | **84.7** | 41.3 | 33.3 | 97.3 | 4.3 | 6.7 | 52.1 |
| LTM-S (Ours) | **77.3** | **95.0** | **39.0** | **20.3** | **82.7** | 84.3 | **52.3** | **68.3** | **98.0** | **10.3** | **7.7** | **57.7** |

training under identical conditions. All other baselines are implemented following settings from Li et al. (2024d). For *System 1*, we construct a hand-crafted mapping function combining ideas from Ko et al. (2023); Li et al. (2024d) (details in Appendix G). We follow evaluation settings identical to Li et al. (2024d), evaluating each policy across 4 task styles with fixed camera view and 20 randomized trials per task style. Each trial uses different initial positions of the objects present in the environment.

**Zero-Shot Results:** We present results for zero-shot evaluation in Table 2. Strong performance of this *pre-trained only* Sys-2 module highlight the significance of our large-scale pretraining.

**Finetuning Results:** We next fine-tune our Sys-2 module on the robot (RD) and human (HD) demonstrations, presenting these results in Table 3. Compared to AVDC (Ko et al., 2023) that makes RGB predictions, our Sys-2 module that predicts pixel motion benefits from human demonstrations (+2.5%). In contrast, AVDC predictions break down when trained on both RD and HD. We attribute this to the greater difference between human vs robot manipulators in RGB space compared to pixel motion space (see motion overlay in Figure 5, distribution analysis in Appendix L, and Xu et al. (2024)). We also highlight that fine-tuning here uses only video-caption pairs and no action ground-truth. This is what allows learning from both RD and HD data. Collection of such human demonstrations (HD) is much faster compared to teleoperated robot demonstrations (RD) (Cheang et al., 2025), underscoring the value of our LangToMo framework.

## 4.2 METAWORLD SIMULATED ENVIRONMENT

Our second set of evaluations use 11 tasks from the MetaWorld (Yu et al., 2019) simulated environment containing a Sawyer robot arm, constructed following Ko et al. (2023). We select this environment and tasks for direct comparison to Ko et al. (2023), which is the closest prior work to our method: they use dense pixel motion for robot manipulation but extract it from generated frame pairs. These tasks also span key challenges in robot control such as complex 3D motions (e.g. button-press-top), contact-rich manipulation (e.g. basket-ball), and semantic understanding (e.g. door open vs close). Each task episode corresponds to successfully completing an action described in natural language.

**Training:** We pretrain *System 2* on the OpenX subset, followed by additional training on 165 MetaWorld videos (identical to the split used in Ko et al. (2023)). For the learned variant of *System 1*, we train on 20 expert demonstrations per task (220 total). We also implement a hand-crafted variant of System 1, following the design in Ko et al. (2023) to ensure fair comparison.

**Baselines:** All baselines follow settings in Ko et al. (2023)). The behaviour cloning (BC) baselines are trained on 15,216 labeled frame-action pairs (over 5x more data). BC-Scratch uses a randomly initialized ResNet-18 while BC-R3M uses pretrained weights from Nair et al. (2022). Diffusion Policy follows settings in Chi et al. (2023) and is trained on the same data. UniPi (Du et al., 2023b) uses the outputs of the AVDC model and its predictor is trained on the same data used for BC baselines.

Table 5: **CALVIN Evaluation:** Zero-shot long-horizon evaluation on the Calvin ABC→D benchmark where agent is asked to complete five chained tasks sequentially based on instructions. More results in Table 8.

| Method | Training Data | $i^{th}$ Task Success Rate ↑ | | | | | Avg. Len ↑ |
|---|---|---|---|---|---|---|---|
| | | 1 | 2 | 3 | 4 | 5 | |
| RT-1 | 100% ABC | 0.533 | 0.222 | 0.094 | 0.038 | 0.013 | 0.90 |
| Diffusion Policy | 100% ABC | 0.402 | 0.123 | 0.026 | 0.008 | 0.00 | 0.56 |
| Robo-Flamingo | 100% ABC | 0.824 | 0.619 | 0.466 | 0.331 | 0.235 | 2.47 |
| Uni-Pi | 100% ABC | 0.560 | 0.160 | 0.080 | 0.080 | 0.040 | 0.92 |
| MDT | 100% ABC | 0.631 | 0.429 | 0.247 | 0.151 | 0.091 | 1.55 |
| Susie | 100% ABC | 0.870 | 0.690 | 0.490 | 0.380 | 0.260 | 2.69 |
| GR-1 | 100% ABC | 0.854 | 0.712 | 0.596 | 0.497 | 0.401 | 3.06 |
| Vidman | 100% ABC | 0.915 | 0.764 | 0.682 | 0.592 | 0.467 | 3.42 |
| RoboUniview | 100% ABC | 0.942 | 0.842 | 0.734 | 0.622 | 0.507 | 3.65 |
| LTM-S (ours) | 100% ABC | 0.971 | 0.824 | 0.728 | 0.672 | 0.606 | 3.81 |
| GR-1 | 10% ABC | 0.672 | 0.371 | 0.198 | 0.108 | 0.069 | 1.41 |
| VPP | 10% ABC | 0.878 | 0.746 | 0.632 | 0.540 | 0.453 | 3.25 |
| LTM-S (ours) | 10% ABC | 0.896 | 0.769 | 0.652 | 0.596 | 0.467 | 3.38 |

AVDC (Flow & Default) are trained identical to Ko et al. (2023) using same 165 Metaworld videos, same for $\pi_0$ baseline but using action ground-truth. AVDC (PT) is additionally pretrained on our OpenX subset making the training identical to LTM. Im2Flow2Act (Xu et al., 2024) and ATM (Wen et al., 2023) follow their default implementations and are also trained identically using the same training data as LangToMo.

**Evaluation:** Following evaluation settings identical to (Ko et al., 2023), we evaluate each policy across 11 tasks. For each task, videos are rendered from 3 distinct camera poses, with 25 randomized trials (different initial positions of the robot arm and objects) for each view.

**Results:** We present the success rates for the 11 tasks and the average across tasks in Table 4. Notably, several prior works (Du et al., 2023b; Ko et al., 2023) exhibit moderate success rates, underscoring the difficulty of the benchmark. VLAs like $\pi_0$ (Intelligence et al., 2025) that require extensive action ground-truth for fine-tuning also suffer under this data constrained evaluation setting. Both our LTM-H and LTM-S variants achieve strong overall performance, highlighting the effectiveness of our framework. Our approach of directly predicting pixel motion compared to RGB in AVDC (Ko et al., 2023) achieves clear performance improvements (+9.0%). Moreover, AVDC fails to benefit from pretraining, which we attribute to the greater domain gap across embodiments in RGB space compared to pixel motion space. Another important point of comparison is the AVDC (flow) baseline, which also uses pixel motion prediction but differs in model architecture, flow representation, and training procedures. We attribute this improved performance of LangToMo over AVDC to our unique design choices. Our improvements in comparison to ATM (Wen et al., 2023) that uses visual traces (+5.7) and Im2Flow2Act (Xu et al., 2024) that uses optical flow subsets (+55.0) demonstrate the usefulness of our dense pixel motion features, which is analogous to future optical flow predictions.

## 4.3 CALVIN SIMULATED ENVIRONMENT

CALVIN (Mees et al., 2021) is another simulation benchmark used in recent works such as Hu et al. (2025). We evaluate our model following settings in Hu et al. (2025) and summarize these results in Table 5. All prior work numbers are directly borrowed from Hu et al. (2025); Nguyen et al. (2025) since we follow their exact settings for evaluation.

**Results:** We explore the two settings of training on the full ABC split and 10% of the ABC split. We outperform multiple prior work on both settings, we strong improvements in the data constrained 10% ABC setting, surpassing even larger models with significant pretraining such as VPP. We provide additional results and discussion on this benchmark in Appendix B.2.

## 4.4 ABLATION STUDIES

We conduct a series of ablative studies with LTM-S on the MetaWorld benchmark to evaluate the importance of key components within LangToMo. Results are summarized in Table 6.

Table 6: **Ablation Study:** We report mean success rate % (overall) on MetaWorld benchmark with our LTM-S variant. (left) Results highlight importance of key components in our System-2 model. (right) Results justify several high-level design choices of our framework.

| Img | Lang | Prev Flow | PT | Overall |
|---|---|---|---|---|
| ✓ | ✓ | ✓ | ✓ | 57.7 |
| ✓ | ✓ | ✓ | ✗ | 53.1 |
| ✓ | ✓ | ✗ | ✗ | 50.2 |
| ✓ | ✗ | ✗ | ✗ | 39.7 |
| ✗ | ✓ | ✗ | ✗ | 5.6 |

| Method | Overall |
|---|---|
| Ours (default) | 57.7 |
| No diffusion | 16.2 |
| CA instead of concat | 15.8 |
| Sys-1 & 2 same freq | 48.7 |
| Only learned Sys-1 | 3.2 |

**System 2 Input Conditioning & Pretraining:** In Table 6 (left), removing visual ("Img"), language ("Lang"), or history information ("Prev Flow") from conditional inputs to the diffusion model significantly reduces performance, highlighting importance of each conditioning signal. On the other hand, removing diffusion model pretraining ("PT") leads to a modest performance drop, indicating that while pretraining aids convergence and performance, the framework remains effective with limited finetuning alone.

**Simpler Baselines:** We explore several alternate design choices in Table 6 (right). Replacing diffusion ("No diffusion") with an autoencoder breaks System-2 learning process. We believe diffusion is more suited for learning the multi-modal output-space of language to motion distributions. Modifying conditioning strategy to cross-attention ("CA instead of concat") also degrades performance. We attribute this to loss of spatial information when performing cross-attention with spatially-averaged visual embeddings. Skipping the iterative System-1 design (running System-1 at same frequency), and generating multiple actions per System-2 generated motion at once ("Sys-1 & 2 same freq") also degrades success rates, validating our design choices. Additionally, bypassing intermediate motion representations ("Only learned Sys-1") leads to poor results, underscoring the clear role played by our System-2 module. See Appendix K for a detailed discussion.

**Inference Costs:** Our current end-to-end LangToMo framework utilizes 25 DDIM iterations of System-2 followed by System-1 resulting in a considering inference time of 3.8 seconds for one action sequence. However, following techniques from prior work (Hu et al., 2025), our inference pipeline can easily be extended to achieve up to $25\times$ inference speedup resulting in runtime comparable to existing VLA works such as VPP (Hu et al., 2025) and $\pi_0$ (Black et al., 2024). We provide further discussion discussion on runtime and efficient inference in Appendix I.

## 5 CONCLUSION

We presented LangToMo, a scalable vision-language-action framework that decouples motion generation and action execution through a dual-system architecture. By leveraging image diffusion models to learn universal pixel motion representations from video-caption data, our *System 2* enables generalizable, interpretable motion planning without human annotated dense supervision. Motion features from our System 2 are translated into robot actions by our embodiment-aware *System 1*, using either learned or hand-crafted mappings. Extensive experiments across simulated and real-world environments demonstrate strong performance of our LangToMo framework, highlighting the promise of dense pixel motion representations as a bridge between language, vision, and action for scalable robot learning.

**Limitations:** LangToMo is pretrained on large-scale, embodiment agnostic video-caption data, but relies on embodiment specific System 1 (similar to most prior works) which can be costly for each new embodiment. Learning robust, transferable mappings remains an open challenge. Also, our framework models motion using 2D pixel motions, which lacks explicit depth cues. Extending to 3D motion representations is left as a future direction. In terms of speed, despite operating at sparse intervals, System 2 relies on iterative reverse diffusion that remains computationally expensive at inference time, limiting use in resource-constrained deployments. This is another future direction we hope to explore further. Finally, we currently do not account for ego motion in training videos: we limit our training to fixed camera videos (no ego motion). A key next direction is extending our System-2 training to include videos with ego motion, which would allow scaling to any kind of video.

**Reproducibility:** Our code and models will be released publicly. We use public datasets for our model pretraining. All evaluations follow prior published work that are publicly available.

ETHICS STATEMENT

Our real world experimentation involves video data collected by human participants in the form of human demonstrations and tele-operated robot demonstrations, similar to prior work. In fact, we follow prior published work in building our experimental setup in the real world. All human participation was voluntary, involved no personally identifying or sensitive information, and followed protocol in prior published work. All our experiments were conducted under controlled laboratory conditions with appropriate safety measures to ensure no risk of harm to participants. The proposed framework is intended solely for advancing research in pixel-motion–based robot control and should not be deployed in safety-critical or harmful applications without additional safeguards.

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

# Appendix

## A DATASET DETAILS

We use a subset of OpenX for pretraining of our System-2 module. We use `fractal`, `taco_play`, `language_table`, `stanford_hydra`, `ucsd_pick_place`, `cmu_pickup`, and `utaustin_mutex` datasets from the OpenX collection. Frame sampling is performed uniformly to maintain a fixed, common action count between frames across each dataset by normalizing for control frequency. We present details of each subset in Table 7.

Table 7: **Pretraining Dataset:** We use 7 sub-datasets from the OpenX collection for pretraining of our System-2 module. Note that training is performed jointly with 3 different embodiments operated at different control frequencies. Our pixel motion based representations allows training jointly with such data using a common training objective across data from all embodiments.

| Dataset | Control Frequency | Episodes | Size (GB) | Robot |
|---|---|---|---|---|
| `fractal` | 3 | 73,499 | 111.06 | Google Robot |
| `taco_play` | 15 | 3,242 | 47.77 | Franka |
| `language_table` | 10 | 442,226 | 399.22 | xArm |
| `stanford_hydra` | 10 | 550 | 72.48 | Franka |
| `ucsd_pick_place` | 3 | 1,355 | 3.53 | xArm |
| `cmu_pickup` | 20 | 520 | 50.29 | Franka |
| `utaustin_mutex` | 20 | 1,500 | 20.79 | Franka |

## B ADDITIONAL EXPERIMENTAL RESULTS

We present more experiments on our real world environment as well as two additional simulation environments to further investigate behaviour of our LangToMo framework.

### B.1 SEMANTIC AWARENESS

In LangToMo, our System-2 module plays the role of understanding semantics within the action goal (textual command) and converting it into meaningful pixel motion representations. We empirically validate this functionality by visualizing two examples that contain the same visual observation but different action goals. We illustrate this in Figure 6. Our LangToMo System-2 module shows much better language awareness in comparison to the AVDC baseline (Ko et al., 2023).

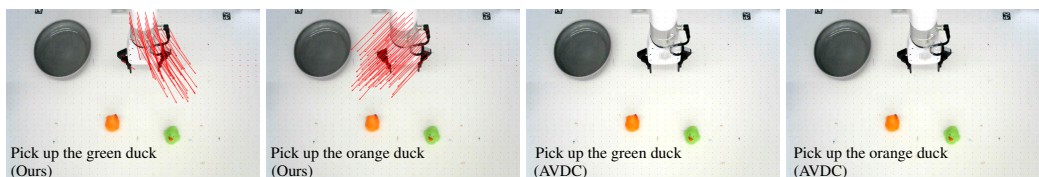

Figure 6: **Semantic Awareness Visualization:** We visualize outputs from our System-2 module (ours; left two figures) for two examples containing the same starting state (visual observation) but different action goals (textual command). LangToMo generates meaningful motions for scenarios needing semantic understanding. We also compare against the AVDC baseline (Ko et al., 2023) (trained on data identical to our LangToMo) that generates next frame RGB images instead of pixel motion. For both cases, AVDC generates the same input frame as its output, i.e. a static next image, seemingly disregarding the language command.

### B.2 CALVIN EVALUATION

CALVIN (Mees et al., 2021) is another simulation benchmark used in several recent works such as Hu et al. (2025). We evaluate our model on this benchmark following settings in Hu et al. (2025) and summarize these results in Table 8. All prior work numbers are directly borrowed from Hu et al.

Table 8: **CALVIN Evaluation:** Zero-shot long-horizon evaluation on the Calvin ABC→D benchmark where agent is asked to complete five chained tasks sequentially based on instructions.

| Method | Training Data | $i^{th}$ Task Success Rate ↑ | | | | | Avg. Len ↑ |
|---|---|---|---|---|---|---|---|
| | | 1 | 2 | 3 | 4 | 5 | |
| RT-1 | 100% ABC | 0.533 | 0.222 | 0.094 | 0.038 | 0.013 | 0.90 |
| Diffusion Policy | 100% ABC | 0.402 | 0.123 | 0.026 | 0.008 | 0.00 | 0.56 |
| Robo-Flamingo | 100% ABC | 0.824 | 0.619 | 0.466 | 0.331 | 0.235 | 2.47 |
| Uni-Pi | 100% ABC | 0.560 | 0.160 | 0.080 | 0.080 | 0.040 | 0.92 |
| MDT | 100% ABC | 0.631 | 0.429 | 0.247 | 0.151 | 0.091 | 1.55 |
| Susie | 100% ABC | 0.870 | 0.690 | 0.490 | 0.380 | 0.260 | 2.69 |
| GR-1 | 100% ABC | 0.854 | 0.712 | 0.596 | 0.497 | 0.401 | 3.06 |
| Vidman | 100% ABC | 0.915 | 0.764 | 0.682 | 0.592 | 0.467 | 3.42 |
| RoboUniview | 100% ABC | 0.942 | 0.842 | 0.734 | 0.622 | 0.507 | 3.65 |
| VPP | 100% ABC | 0.965 | 0.909 | 0.866 | 0.820 | 0.769 | 4.33 |
| DreamVLA | 100% ABC | 0.982 | 0.946 | 0.895 | 0.834 | 0.781 | 4.44 |
| LTM-S (ours) | 100% ABC | 0.971 | 0.824 | 0.728 | 0.672 | 0.606 | 3.81 |
| GR-1 | 10% ABC | 0.672 | 0.371 | 0.198 | 0.108 | 0.069 | 1.41 |
| VPP | 10% ABC | 0.878 | 0.746 | 0.632 | 0.540 | 0.453 | 3.25 |
| LTM-S (ours) | 10% ABC | 0.896 | 0.769 | 0.652 | 0.596 | 0.467 | 3.38 |

(2025) since we follow their exact settings for evaluation. We explore the two settings of training on the full ABC split and 10% of the ABC split. Evaluation is always performed on the unseen D split. Each task is a set of five sequential sub-tasks and we use the task success rates along with average length metrics for evaluation similar to Hu et al. (2025).

In the first case (100% ABC), we perform competitively outperforming several recent works. Concurrent works, VPP (Hu et al., 2025) and DreamVLA (Zhang et al., 2025) outperform us on this split. We note that both these models are pretrained on significantly more data than our LTM model. VPP also uses a larger sized model (1.5B) compared to our LTM (0.86B). Moreover, unlike in the real world, our method cannot directly benefit from human videos for these simulated environments. Nevertheless, in the second case (10% ABC), we outperform VPP and GR-1 (Wu et al., 2023) highlighting the data efficiency aspect of our LangToMo framework.

## B.3 iThor Evaluation

We next explore the ability to extend our method to benchmarks that involve ego motion of the robot (e.g. simple navigation tasks). Following prior work AVDC (Ko et al., 2023), we evaluate on the iThor benchmark and present results in Table 9. Results indite clear improvements of our proposed LangToMo over naive baselines and prior work AVDC (Ko et al., 2023). The behaviour cloning (BC) baselines are implemented following Ko et al. (2023). Both AVDC and LTM-H (ours) are trained on the same data under common training settings for fair comparison.

Table 9: **Results on iThor Benchmark:** We follow the iThor dataset based evalution setup used in AVDC (Ko et al., 2023) to demonstrate that our method generalizes to robot movement based control as well (i.e. where ego motion occurs). Results indicate that our method outperforms AVDC across categories and overall.

| Method | Kitchen | Living Room | Bedroom | Bathroom | Overall |
|---|---|---|---|---|---|
| BC-Scratch | 1.7 | 3.3 | 1.7 | 1.7 | 2.1 |
| BC-R3M | 0.0 | 0.0 | 1.7 | 0.0 | 0.4 |
| AVDC | 26.7 | 23.3 | 38.3 | 36.7 | 31.3 |
| LTM-H (ours) | 27.3 | 24.3 | 40.0 | 36.7 | 32.2 |

Our LangToMo was not explicitly designed for such ego-motion tasks, but nevertheless is capable of performing such tasks similar to AVDC. We take these results as a promising indication that LangToMo can be further extended to better handle such ego-motion tasks.

## C  RELATIVE PIXEL MOTION

A key design choice in our formulation is to represent pixel motion with respect to the current frame ($\boldsymbol{x}_t$), rather than the previous frame ($\boldsymbol{x}_{t+1}$) or some other frame. This aligns with the structure of our conditional diffusion model, which receives $\boldsymbol{x}_t$ as a secondary conditioning input. Predicting the transformation from $\boldsymbol{x}_t$ to the next frame allows the model to more directly focus on the visual cues present in the current state. In contrast, predicting motion from $\boldsymbol{x}_{t-1}$ or some other different frame would require indirect reasoning over a non-visible state, introducing additional complexity. Hence our approach is to represent past pixel motion (e.g. $\boldsymbol{x}_{t-1}$ to $\boldsymbol{x}_t$) as $\boldsymbol{x}_t$ to $\boldsymbol{x}_{t-1}$ instead. While this may seem counterintuitive, we note how prior literature on image-pair-based optical flow prediction for video tasks has also found that defining motion in terms of a reference image—particularly the current frame that is visible—can lead to more stable and accurate flow estimates (Liang et al., 2024a). Moreover, our experiments representing previous motion in a different manner lead to subpar performance, standing as further evidence.

We also experiment trying to predict an additional future motion relative to a future frame. We compare this against predicting that same future motion relative to the current frames. In this setting, the latter performs well while the former variant fails to learn meaningful motion signals predictions.

## D  LANGUAGE EMBEDDING MODEL

For the language embedding model, we employ the Universal Sentence Encoder (USE), a pre-trained model from (Cer et al., 2018). USE generates fixed-length vector representations of text, capturing rich semantic meaning, making it suitable for various natural language processing (NLP) tasks. Its widespread use in research, including works like OpenX (Padalkar et al., 2023), highlights its effectiveness in transforming textual input into meaningful embeddings even for robotic tasks. In our framework, the USE serves as a key component, encoding language instructions into dense vectors that are later used to guide the generation of motion representations. The model's ability to produce consistent and high-quality embeddings enables seamless integration between language and vision modalities, ensuring that our system can accurately interpret and respond to diverse language commands.

## E  DIFFUSION MODEL DETAILS

In our diffusion model training, input noising is applied by adding Gaussian noise to the target motion data (following standard settings from Ho et al. (2020)). The image condition input and the previous flow are not subject to this noising. The previous flow is corrupted with a 50% chance. During corruption, a random amount of Gaussian noise is added. To ensure diverse and meaningful training, filtering and augmentation operations are performed on the frames as described next. The indices corresponding to consecutive frames ($i$ and $i + 1$) are selected such that they maintain fixed intervals based on the video frame rate. Frames with zero optical flow (i.e., no motion) between $i$ and $i + 1$ are filtered out to avoid irrelevant data. Additionally, to handle the completion of textual instructions, we introduce zero motion at the ends of videos, ensuring that these states map to a lack of motion when the instruction concludes. The visual inputs (images and optical flow) are cropped and resized, with appropriate transformations applied to the flow data to maintain consistency.

## F  HAND-CRAFTED MAPPING FUNCTIONS

Pixel motion (and optical flow) is meaningful as standalone features. For $x \in \mathbf{R}^{2 \times H \times W}$, each $x_{0,h,w}$ and $x_{1,h,w}$ denotes the $\Delta x$ and $\Delta y$ pixel motion for spatial location $(x, y)$. This allows humans to visually understand the meaning of any pixel motion feature, and also connect it to hand-crafted algorithms for further exploration, adding an increased level of interpretability to our work.

**Synthetic Environments:** We follow the formulation of Ko et al. (2023) using a segmentation map of robot controller and a depth map of environment. The generated pixel motions are converted into directions in 3D space to move the robot controller based on these dense maps. We direct the reader to Ko et al. (2023) for further details.

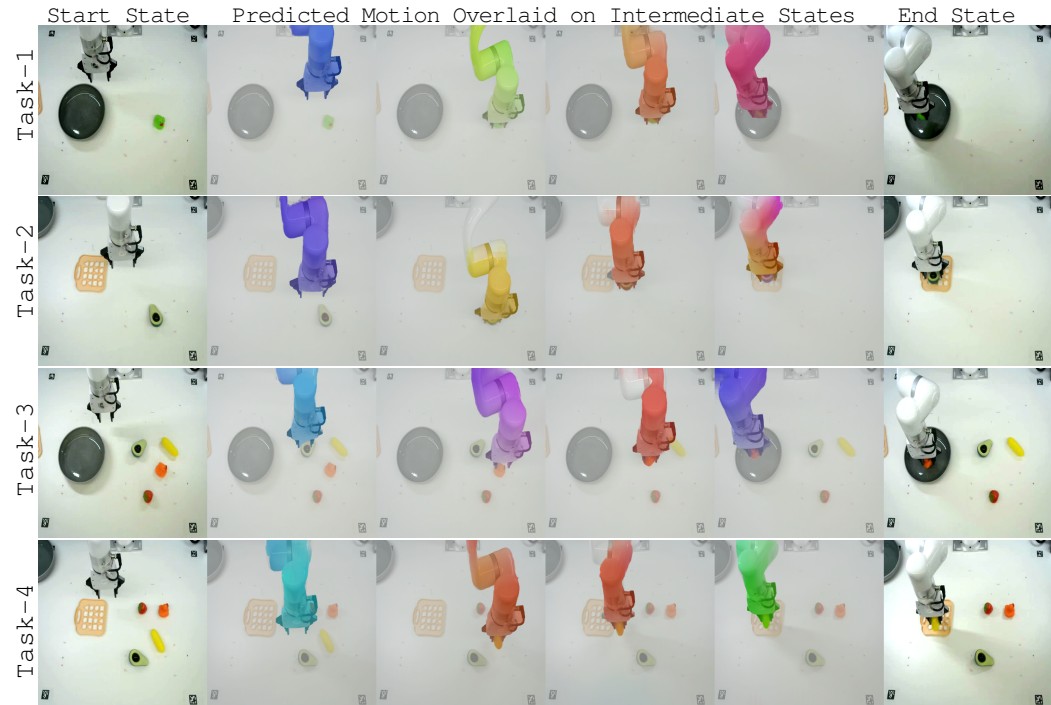

Figure 7: **Real World Tasks:** We illustrate the four real-world tasks following LLaRA Li et al. (2024d). Start and end states are shown in the first and last columns, with predicted pixel motion (color indicates motion direction) overlaid on intermediate states. LangToMo performs these challenging tasks successfully (see results in Table 3).

**Real World Environments:** Motivated by Li et al. (2024d), we build our real world environment with a single plane assumption (e.g. table top manipulation) and map the predicted pixel motions for the robot controller center points onto the plane (using visual geometry). An initial camera calibration is performed for the environment to obtain necessary camera matrices. After extracting a start and end position for a manipulation task following this setting, our position to action vector conversion is identical to Li et al. (2024d). Examples of these tasks are shown in Figure 7.

## G  REAL WORLD EXPERIMENTS

We perform four styles of real world experiments as illustrated in Figure 7. The language instructions for the four examples in this figure, where each belongs to one of the four task styles, are as follows:

1. `Pick up the duck and place on the bowl.`
2. `Pick up the duck and place on the tray.`
3. `Pick up the avocado and place on the bowl.`
4. `Pick up the corn and place on the tray.`

Each task style contains similar textual commands that require some object manipulation in the table top environment. We select these following Li et al. (2024d) to ensure fair comparisons to prior work.

For real world experiments, our System-2 is called every 2 seconds, with System-1 being called every 0.2 seconds.

## H  INFERENCE PIPELINE

Our inference procedure operates as a hierarchical control loop, integrating our high-level System 2 with the low-level System 1. The process begins with an initial objective generated from System 2.

Table 10: **Runtime Analysis:** We compare the runtime of our method against several prior works. We also highlight our inference efficient variant constructed following prior work VPP (Hu et al., 2025).

| Method | Denoising Steps | Runtime | CALVIN (ABC 10%) | MetaWorld |
|---|---|---|---|---|
| Ours (default) | 25 | 3.8s | 3.38 | 57.7 |
| Ours (variant) | 1 | 0.16s | 3.36 | 57.1 |
| VPP | 1 | 0.21s | 3.25 | - |
| $\pi_0$ | - | 0.59s | - | 2.7 |

System 1 handles the fine-grained control for this objective, and its execution follows a defined termination protocol with two primary conditions: it runs for a maximum of 10 control steps, or it converges to a predefined state in hand-crafted settings. When either condition is met, System 1's execution for that sub-goal terminates, and the process immediately loops back to re-invoke System 2. System 2 then provides a new high-level objective, which is passed to System 1 to continue the task.

The overall task is governed by a total episode step limit. If this limit is reached before the task is successfully completed, the entire episode is considered a failure. Success is determined based on the environment: in a simulated environment, an episode is deemed successful when a predefined task success signal is received; in a real-world setting, success is confirmed by a human tester. Upon success, execution is halted, and the episode is concluded.

## I  Inference Efficiency

We discuss the inference runtime costs of our method and compare against prior work in Table 10. Motivated by the single diffusion denoising step based robot control pipeline in VPP Hu et al. (2025), we construct a modified inference pipeline to our LangToMo, which similarly uses a single denoising step in the System-2 diffusion model during inference on downstream tasks. Our System-2 model training remains unchanged while the System-1 model necessitates joint fine-tuning with our System-2 outputs. While this introduces this additional System-1 finetuning phase to our training pipeline, this variant results in over $25\times$ speedup in our inference compared to our default inference pipeline that utilized 25 DDIM iterations.

We provide our measurements of runtime as well as performance from this modified inference pipeline in Table 10. Results are also reported for baselines VPP (Hu et al., 2025) and $\pi_0$ (Black et al., 2024). While our default LangToMo framework achieves stronger performance on two tasks at the cost of lagging behind these VLAs in runtime, the faster-runtime variant achieves almost similar performance (negligible performance drop across both benchmarks) at a fraction of the original runtime. The limitation of this variant is the additional finetuning of System-1 and the loss of our fully actionless inference capability. However, this additional finetuning is a one-time cost, which we believe is clearly justified by the over 25x speedup in runtime.

In summary, we highlight how in cases requiring faster inference, this variant achieving negligible performance drops across benchmarks is an ideal solution.

## J  Baseline Details

Our key baselines are from AVDC (Ko et al., 2023) and LLaRA (Li et al., 2024d). For both methods, we use their official implementations to replicate their results and evaluate ours under identical settings. For LLaRA, all results are reported on their inBC variant for fair comparison against our method (i.e. similar inputs during inference / no external scene object information). We also use official implementations for VPP (Hu et al., 2025), Im2Flow2Act (Xu et al., 2024), and ATM (Wen et al., 2023) for evaluating those baselines. All these baselines are trained on the same data as our LangToMo model.

## K  Detailed Ablations

We discuss our ablations in Table 6 in detail in the following section.

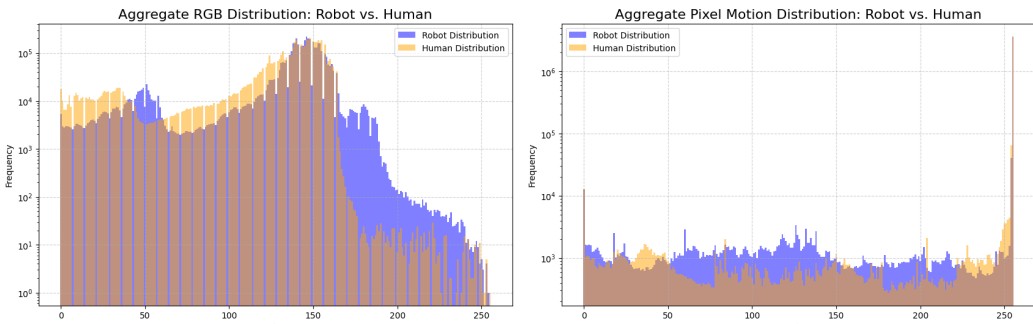

Figure 8: **Histogram Comparisons for RGB and Pixel Motion Distributions of Robot vs Human Demonstrations:** We illustrate two histograms that compare the aggregate pixel value distributions of 40 human and 40 robot demonstrations. For the RGB distributions (left), the high degree of separation and low overlap between the distributions of the two groups indicate significant differences in appearance, resulting in a high Symmetrized KL Divergence of 0.7881. In contrast, the pixel motion distributions (right) show substantial overlap between RGB and pixel motion, demonstrating the strong similarity in kinematic patterns between human and robot demonstrations, leading to a much lower Symmetrized KL Divergence of 0.0199. These results suggest that pixel motion is a more embodiment-agnostic metric for comparing demonstrations.

**System 2 Design Choices:** We first ablate critical inputs to *System 2 (Motion Generation)*. Removing pretraining ("PT") leads to a modest performance drop (from 53.6% to 53.1%), indicating that while pretraining aids convergence, the framework remains effective with limited finetuning alone. Removing the previous optical flow input ("Prev Flow") results in a larger decline to 50.2%, validating the importance of temporal conditioning. Ablating the language embedding leads to a significant drop (to 39.7%), highlighting the necessity of semantic instruction guidance. Finally, removing the visual input ("Img") results in near-random performance (5.6%), confirming that visual grounding is essential.

**High-Level Framework Design:** We next evaluate several higher-level architectural decisions. Removing the diffusion model ("No diffusion") and training a direct regression using an autoencoder based setup (using same architecture as our diffusion model but without noise inputs and with a single time-step for training and inference) leads to a sharp performance drop (to 16.2%), underscoring the value of iterative, probabilistic modeling for motion generation. Replacing input concatenation with cross-attention ("CA instead of concat") similarly degrades performance, suggesting that simple spatial concatenation is a more effective conditioning strategy for our setting. Using a multi-action decoder within *System 1* to run it at same frequency as our system 2 ("Sys-1 & 2 same freq") results in slightly lower performance (48.7%), indicating that our default action mapping is more effective. Training only a learned *System 1* without leveraging pixel motions generated by Sys-2 ("Only learned Sys-1") performs poorly (3.2%), demonstrating that our System-1 module simply learns to map the generated pixel motion to robot manipulator actions.

## L  PIXEL MOTION DISTRIBUTION ANALYSIS

We present an analysis of the divergence between human and robot demonstration data using both RGB pixel values and pixel motion. To quantify the difference, we first aggregated the pixel value distributions from 40 human and 40 robot demonstrations, creating two distinct distributions for each data type. We then computed the Symmetrized Kullback-Leibler (KL) divergence to measure the difference between the human and robot distributions. The results, with a high KL divergence for RGB (0.7881) and a very low one for pixel motion (0.0199), indicate that the distributions of RGB pixel values are significantly different between human and robot demonstrations, while the distributions of pixel motion are remarkably similar. As illustrated in Figure 9, this finding supports our hypothesis that pixel motion is a more embodiment-agnostic representation. The high divergence in RGB is a function of embodiment-specific factors such as lighting, skin tone, robot color, and background, which vary greatly between human and robotic forms. Conversely, the low divergence in pixel motion demonstrates that, regardless of the physical embodiment, the fundamental kinematic

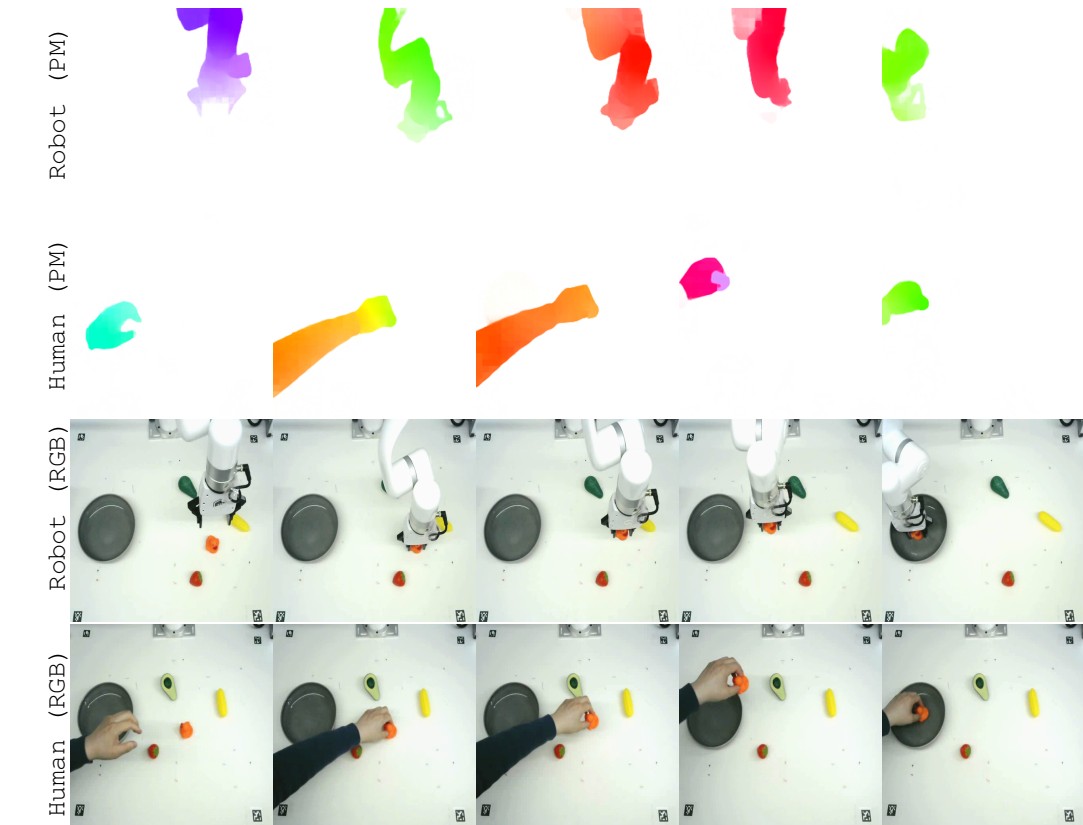

Figure 9: **Visualization of Pixel Motion Space vs RGB Space:** The comparative difference between demonstrations of different embodiments, even human vs robot, are much lower in pixel motion (PM) space (top) compared to RGB space (below). In this example, we visualize observations of 5 consecutive states for two demonstrations from robot and human embodiments to highlight qualitative results supporting our claim. The PM visualization uses HSV color space where H denotes the flow direction and S-V channels denote the flow magnitude. Notice the common colors (motion direction) between PM pairs in the last 3 columns as well as similar textures / color intensity between all PM pairs (in comparison to RGB pairs).

patterns of the motion itself are consistent, making it a more universal measure for learning from demonstrations.

## M EXTENDED RELATED WORK

In this section, we discuss some recent related works in more detail. The idea of using pixel motions or optical flows for robot control tasks has gained significant interest recently. Several seminal works explore and establish the promise of this direction.

In Yuan et al. (2024a), a VAE conditioned on language is trained to predict sequences of points given 3D RGB-D inputs. Their model uses depth-maps for both training and inference. They focus their trajectories on object regions, avoiding manipulator motion regions. The trajectories are generated once, following a close-loop style control. They only explore manual robot policies that map the trajectory predictions to robot actions.

On the other hand, Gao et al. (2025) uses optical flow as a way of improving video generation for robot control. In their framework, flow conditioned RGB videos are first generated. The target futures contained in RGB frames of videos are used to calculate value functions that control a robot policy. Their framework uses a transformer based VAE to generate optical flows for a sparse-subset of points on a 2D grid over the image. The generated flows do not directly control robot manipulation, but are rather used to generate a video; the video generation component is considered as their dynamics module. The video frames are used to construct values functions as opposed to features ingested by a later module.

ATM (Wen et al., 2023) is a manipulator motion focused method using point trajectories for robot control with a dual system setup. They model tracks as 1D sequences of points with a transformer module. A secondary policy transformer is trained to map these point sequences to robot actions. Their trajectory generation is conditioned on the current observation and language command only, with no history conditions. The points are also sparsely sampled, but account for both manipulator and object movements. In their track transformer, each point is separately encoded into one token, overlooking any explicit awareness on their 2D structure. Their ATM framework *"is permutation invariant to the input set of points"*, highlighting the explicit design choice to avoid accounting the structured ordering of pixels. This contrasts with the explicitly 2D structured dense pixel motion used in our LangToMo framework.

A similar approach is explored in Bharadhwaj et al. (2024b) where goal images are used instead of a textual condition. Point tracks for a randomly selected subset of pixels are generated using a diffusion transformer. The selected pixel subset does not occupy a standard 2D grid at training. At inference, they also utilize the goal image, allowing the model to identify which pixels have moved. Optionally, specific pixels to track are also provided during inference, which can be calculated using the goal image.

In Bharadhwaj et al. (2024a), an extended framework is explored where a video generation model, conditioned on current observation and textual goal, first generates future frames. These generated future frames are used to calculate point tracks, which are in turn converted to robot actions. This work resembles somewhat an opposite of Gao et al. (2025); the first generates videos to then generate better point tracks while the latter first generates point tracks to generate better videos.

Im2Flow2Act proposed in Xu et al. (2024) builds a network for flow generation that focuses on object movements avoiding manipulator motion regions. The generated pixel trajectories are processed by a secondary trained module to output robot actions. During both training and inference, they rely on the textual command and external object localization tools to select a pixel subset whose trajectories are generated by their trained network. The flow is generated once per inference episode, operating in a closed-loop style manner.

Zhi et al. (2025) extends the ideas behind Im2Flow2Act to account for depth, calculating pixel motion in 3D. Their pixel trajectories also focus only on object movement (avoids manipulator movement). For inference, the generated object movement trajectories are used to calculate robot action manually, using the 3D coordinates with robot state and geometric operations. Their pixel trajectories are also generated once per inference episode, operating in a closed-loop style manner.

A different approach taken in Hu et al. (2025) involves a video diffusion model trained for future prediction that is used as a feature extractor (as opposed to generating videos) during inference. A secondary trained module maps these diffusion model features into robot actions. Zhong et al. (2025) use optical flow to improve video generation, similar to Gao et al. (2025) but operates autoregressively, generating per-frame motions followed by next frame and so forth. They also use an LLM style transformer for visual generation, using discretized tokens to represent images or motion represented as RGB. In Zhang et al. (2025), a similar LLM style transformer is used to generate robot actions direction, but is trained on several auxilliary tasks, including future prediction for the motion heavy regions of a given video.

Furthermore, multiple recent works have highlighted the significant potential of learning from human or robot videos to acquire skills and representations for robotics tasks, enabling robots to perform complex manipulations with minimal or no explicit hand-crafted algorithms (Bahl et al., 2022; Kerr et al., 2024; Pathak et al., 2018; Valassakis et al., 2022; Wang et al., 2023a; 2024; Argus et al., 2020b; Vecerik et al., 2024; Zhu et al., 2024; Barcellona et al., 2024; Chang et al., 2020; Ponimatkin et al., 2025; Ren et al., 2025b; Sermanet et al., 2018; Zhou et al., 2025a; Ranasinghe et al., 2024; Bahl et al., 2023; Ju et al., 2024; Karamcheti et al., 2023; Li et al., 2024a;b; Mendonca et al., 2023; Srirama et al., 2024; Liang et al., 2024b; Sun et al., 2024; Ajay et al., 2023; Du et al., 2023c; Patel et al., 2025; Albaba et al., 2025). These works highlight the promise of extracting information like motion trajectories, object affordances, or even high-level plans directly from unstructured video data, which can in-turn benefit robotics tasks. Several other contemporary works also explore different forms of pixel trajectories as intermediate features or auxilliary predictions for robot control tasks (Li et al., 2025b; Song et al., 2025a; Ji et al., 2025; Li et al., 2025a). The use of large-scale video data to learn

generating of future states in different forms is also explored in several recent works (Patel et al., 2025; Zhao et al., 2025; Yao et al., 2025; Zhou et al., 2025b; Pan et al., 2025; Chen et al., 2025a).

In summary, pixel motion has been successfully used to improve video generation with generated videos subsequently being used as visual representations for robot control (Gao et al., 2025; Zhong et al., 2025). In contrast, our LangToMo directly uses pixel motion predictions for robot control, demonstrating its better data efficiency, domain generality, and real-world performance. Ko et al. (2023); Bharadhwaj et al. (2024a) map pixel motion to robot actions, but take a longer route by first generating RGB videos, and then using those frames to calculate pixel motion. Our direct pixel motion generation offers a more straightforward solution that achieves better data efficiency and performance. Several works explore directly using pixel motion as a representation for a secondary module mapping them to robot actions (Yuan et al., 2024a; Bharadhwaj et al., 2024b; Zhi et al., 2025) but limit their focus to object motion, ignoring important manipulator movement information. In contrast, our LangToMo learns to predict dense pixel motion, accounting for both object and manipulator movements. Furthermore, our LangToMo is a first to establish a clear demarcation of System-1 and System-2 modules that are trained independently and also called at different frequencies at inference. These provide more compute efficient training as well as faster inference.

