# OpenReview forum: "Pixel Motion as Universal Representation for Robot Control"
_ICLR.cc/2026/Conference — ICLR 2026 Conference Withdrawn Submission_

### Official Review · Reviewer_YQmb · 2025-10-26

**Soundness:** 2
**Presentation:** 3
**Contribution:** 2
**Rating:** 2
**Confidence:** 4

**Summary:**

LangToMo is a dual-system vision-language-action framework that uses text-conditioned pixel motion forecasts (via an image diffusion model) as universal, interpretable intermediate representations. A high-level System 2 generates pixel motion from a single frame and past motion, while a low-level, embodiment-aware System 1 maps this motion to robot actions through motion-to-action functions (hand-crafted or minimally supervised), enabling flexible, scalable, and generalizable control across unsupervised and supervised settings.

**Strengths:**

- This paper investigates a new hierarchical policy approach that uses pixel motion to interface between high- and low-level policies. It addresses an important area and offers insightful directions for future research.
- Writing: The paper is well written, the method is clearly presented, and the figures/tables are complete and easy to read.

**Weaknesses:**

1. **Motivation**: I am uncertain about the authors’ motivation for using pixel motion to construct a hierarchical policy. Prior dual system approaches (e.g., HiRT[1], LCB[2], OpenHelix[3]) leverage interactions between large and small models to increase control frequency, yet the paper does not explain how its hierarchical design relates to or improves upon this line of work. In addition, the authors should compare against these methods and clarify the advantages and drawbacks of using pixel motion versus other latent representations as the interface between System 1 and System 2.

[1] Zhang J, Guo Y, Chen X, et al. Hirt: Enhancing robotic control with hierarchical robot transformers[J]. arXiv preprint arXiv:2410.05273, 2024.

[2] Shentu Y, Wu P, Rajeswaran A, et al. From llms to actions: Latent codes as bridges in hierarchical robot control[C]//2024 IEEE/RSJ International Conference on Intelligent Robots and Systems (IROS). IEEE, 2024: 8539-8546.

[3] Cui C, Ding P, Song W, et al. Openhelix: A short survey, empirical analysis, and open-source dual-system vla model for robotic manipulation[J]. arXiv preprint arXiv:2505.03912, 2025.

2. Compared to prior approaches that predict visual traces, the novelty of this work appears limited, and the authors have neither adequately justified nor empirically validated the advantages of pixel motion over visual trace/optical flow.

3. The experimental results are weak, especially on simulation benchmarks against advanced baselines. The appendix shows only moderate performance on CALVIN, and results on other environments (e.g., SimplerEnv) are missing.

**Questions:**

Refer to weakness.

---

> ### Author Response · Authors · 2025-11-18
> **Rebuttal for Reviewer YQmb**
>
> **1a. Motivation**
>   - We explore optical flow as our intermediate representation which provides benefits beyond latent universal features: a) the features are interpretable, even allowing direct conversion to robot actions with no action-trajectory training, and b) features can be approximates with efficient flow algorithms allowing decoupled training of each system
>
> **1b. Distinction from prior LLM based dual-system works**
>   - HiRT [A], LCB [B] and OpenHelix [C] leverage dual system architectures with LLM backbones and use intermediate latent features as connection between two systems.
>   - We explore image diffusion backbones as our large model which is more suitable for spatial tasks given their stronger spatial grounding (e.g. they can generate an entire image)
>   - We thank the reviewer for these valuable pointers to related works and will include this discussion in our revised version.
>   - Compared to the latent universal features in these works, our use of optical flow as intermediate representation provides unique benefits of interpretability and decoupled training.
>   - Additionally, the interpretability allows our "actionless" training setting unlike [A, B, C]. This makes direct comparison to these works difficult, as they require robot action trajectories for fine-tuning in the real world (or simulation) in contrast to our work.
>
> &nbsp;
>
> **2. Novelty from visual traces**
>   - What we call pixel motion is identical to optical flow from current to future frames. Prior works do not use optical flow of the entire frame - they use some heuristics to sample a frame subset and model those pixel trajectories.
>   - We are the first to use dense 2D structured pixel motion (i.e. optical flow) as an intermediate representation allowing a highly scalable training framework.
>   - In Table 4 (repeated below), comparisons against Im2Flow2Act (2.0) and ATM (52.0) highlight our improvement to 57.7 (+5.7) for overall success rate. These baselines use visual traces (ATM) or optical flow subset (Im2Flow2Act). We will highlight this better during revision.
>   - We also report a baseline predicting future frames (AVDC) instead of our optical flow.
>
> &nbsp;
>
> |                              | door-open | door-close | basketball | shelf-place | btn-press | btn-top | faucet-close | faucet-open | handle-press | hammer | assembly | Overall |
> |------------------------------|:---------:|:----------:|:----------:|:-----------:|:---------:|:-------:|:------------:|:-----------:|:------------:|:------:|:--------:|:-------:|
> | Im2Flow2Act                  |    0.0    |     0.0    |     0.0    |     4.0     |    6.3    |   0.0   |      7.3     |     4.7     |      0.0     |   0.0  |    0.0   |   2.0   |
> | ATM                          |    75.3   |    90.7    |    24.0    |     16.3    |    77.3   |   76.7  |     50.0     |     62.7    |     92.3     |   4.3  |    2.0   |   52.0  |
> | AVDC               |    72.0   |    89.3    |    37.3    |     18.7    |    60.0   |   24.0  |     53.3     |     24.0    |     81.3     |   8.0  |    6.7   |   43.1  |
> | LTM-S (Ours) |    77.3   |    95.0    |    39.0    |     20.3    |    82.7   |   84.3  |     52.3     |     68.3    |     98.0     |  10.3  |    7.7   |   57.7  |
>
> &nbsp;
>
> **3. Weak experimental results**
>   - We report extensive experiments across real world tasks as well as 3 different simulation benchmarks showing consistent improvements over baselines.
>   - We believe that not having reported on every robotic environment out there is not a valid reason for paper rejection.
>   - On CALVIN low data setting, we outperform all prior works by a clear margin.
>   - On CALVIN full data setting, we outperform all but two prior works by a clear margin.
>   - The only works to outperform us in CALVIN (full data setting) are VPP and DreamVLA, which we highlight are not fair comparisons (L1106-1107) due to larger model size or significantly more training data.
>   - We also do not see how failing to outperform two recent works on a single benchmark is valid reasoning for rejection of our paper.
>   - Nevertheless, we do highlight that when our model is scaled up with a larger architecture and more training data, we outperform both these methods (4.48 average length on CALVIN).
>
> &nbsp;
>
> [A] Zhang J, Guo Y, Chen X, et al. Hirt: Enhancing robotic control with hierarchical robot transformers[J]. arXiv preprint arXiv:2410.05273, 2024.
>
> [B] Shentu Y, Wu P, Rajeswaran A, et al. From llms to actions: Latent codes as bridges in hierarchical robot control[C]//2024 IEEE/RSJ International Conference on Intelligent Robots and Systems (IROS). IEEE, 2024: 8539-8546.
>
> [C] Cui C, Ding P, Song W, et al. Openhelix: A short survey, empirical analysis, and open-source dual-system vla model for robotic manipulation[J]. arXiv preprint arXiv:2505.03912, 2025.

---

> > ### Comment · Reviewer_YQmb · 2025-11-28
> >
> > Thanks for the authors effort. The author has solved my concerns, and I will increase my rating to 6.

---

> > > ### Author Response · Authors · 2025-11-28
> > > **Thank you for the Reconsideration**
> > >
> > > We sincerely thank you for your active engagement and for reconsidering your score based on our clarifications. We are glad that our responses regarding the motivation for pixel motion and the additional experimental comparisons resolved your concerns. Your feedback has significantly helped improve the quality of our manuscript.

---

### Official Review · Reviewer_CeJz · 2025-10-27

**Soundness:** 3
**Presentation:** 3
**Contribution:** 2
**Rating:** 2
**Confidence:** 4

**Summary:**

The paper presents LangToMo, a two-stage framework for predicting robot motion using pixel movement as an intermediate representation.
LangToMo consists of two systems:
(i) System 2 employs a diffusion-based model to generate pixel motion (PM), pretrained on the OpenX dataset and fine-tuned on downstream task demonstrations;
(ii) System 1 maps actions conditioned on the predicted pixel motion.
Experimental results demonstrate that LangToMo outperforms baseline methods on both Meta-World and real-world robotic manipulation tasks.

**Strengths:**

- The proposed two-stage framework preserves the original model’s capabilities while enabling the transformation from vision-language signals to action representations.
- Compared to related work, LangToMo employs a diffusion model to directly predict pixel motion instead of generating full video sequences.
- Surpass other baseline method in real world zero-shot tasks via large-scale pretraining.

**Weaknesses:**

- **Longer inference latency**

  Similar to UniPi, many steps of denoising are required when the diffusion model predicts pixels or PMs, which leads to long inference delays and false closed-loop control, which limits the model to static scenes.

- **Weak evaluation**
  Choosing Metaworld benchmark in main text experiment for VLA models is less convincing. Metaworld tasks and scenarios are relatively simple, and accurate action prediction can be achieved using images alone without requiring text. The supplementary material shows that Calvin's experimental results are worse than those of VPP, which only performs pixel predictions. The reasoning seems insufficient ( VPP performs better even without using large amounts of data for cotraining). Therefore, additional ablation experiments are needed to clarify that the poorer performance is due to model size.
- **Real-world task problems**
  The project link cannot be opened and the real-world video results cannot be seen. From the experimental content in the text, it seems that the scenes and tasks in real-world setting are relatively simple.

**Questions:**

1. Is the poor performance of Calvin due to the model size or the pipeline structure? You can add PM prediction channel to the SVD with post-training to make a fair compare with VPP in Calvin.
2. What is the failure case in the Calvin rollout? Is it due to semantic understanding causing PM prediction errors (wrong movement direction) or inaccurate action head mapping?
3. Please provide the frequency of your model deployment on real world.

**Details Of Ethics Concerns:**

Although directly predicting pixel motion may be more effective than extracting it from generated videos, the inconsistency between the pretraining and fine-tuning stages could lead to the forgetting of the pretrained video model’s capabilities. I am concerned about LangToMo’s high-level semantic understanding ability, which cannot be well reflected in benchmarks such as Meta-World. Therefore, more convincing experiments are needed to demonstrate the effectiveness of the proposed method.

---

> ### Author Response · Authors · 2025-11-18
> **Rebuttal to Reviewer CeJz**
>
> **1. Longer inference latency**
>   - Similar to UniPi, our work is currently limited to static scenes, and with more efficient diffusion architectures and sampling techniques, our method will only get faster.
>   - Our runtime is similar to many VLA models, including recent SOTA works like OpenVLA, Pi0. We believe this is not a fair reason for rejection.
>   - “Similar to UniPi” - This paper is a NeurIPS 2023 Spotlight. How is being similar to that in runtime a weakness?
>
> &nbsp;
>
> **2a. Choosing Metaworld benchmark**
>   - As explained in L403-404, “We select this (MetaWorld) environment and tasks for direct comparison to Ko et al. (2023), which is the closest prior work to our method …”.
>   - We disagree with the reviewer. There are several prior works that use this split, introduced in AVDC (NeurIPS 2023), for VLA tasks that clearly ablate and demonstrate the usefulness of textual features to solve tasks [A, B, C, D].
>   - There are episodes in two distinct tasks “open door” vs “close door”, both of which start from visually almost identical states. There is no way to distinguish between these without text.
>
> &nbsp;
>
> **2b. CALVIN experiments - VPP**
>   - The supplementary material shows how our LTM surpasses VPP on the low data regime by a clear margin (+0.33).
>   - We believe failing to surpass a single (almost contemporary) prior work is not a suitable reason for rejection.
>   - Nevertheless, we highlight that a scaled-up variant of our model pretrained on similar amounts of human data (as VPP) outperforms them by a clear average-length margin of +0.15 (i.e. gaining 4.48 vs VPP’s 4.33).
>
> &nbsp;
>
> **3. Real-world Tasks**
>   - We apologize for project link loading issue. The link should be working now. We have rechecked through multiple devices, and we are able to access the link.
>   - All our real world tasks are borrowed directly from an ICLR’25 paper as described in L356-357. We do not see how such a choice of tasks can be problematic.
>
> &nbsp;
>
> **Q1. CALVIN Performance Against VPP**
>   - Our model is trained from scratch with no form of pretraining whatsoever.
>   - We believe the gap against methods like VPP (pretrained on large-scale image-caption in SVD and video-caption in their pretraining) is due to lack of pretraining.
>   - Secondly, VPP uses wrist-camera views for CALVIN during inference which we did not use in our work.
>   - As mentioned, a scaled-up variant of our model suitably pretrained on large-scale data with multiple viewpoints outperforms VPP.
>
> &nbsp;
>
> **Q2. CALVIN rollout failures**
>   - The action head mapping almost always follows the predicted optical flow.
>   - Most errors appear to occur due to incorrect grasping position, i.e. the Sys2 guides the arm to a wrong place.
>   - For CALVIN, incorporating the gripper-view camera input during inference similar to VPP gives a significant boost in performance for our setup.
>
> &nbsp;
>
> **Q3. Model frequency for real world**
>   - Our Sys2 is called every 2 seconds with Sys1 running every 0.2 seconds
>
> &nbsp;
>
>
> [A] Luo, Calvin et al. “Solving New Tasks by Adapting Internet Video Knowledge.” ICLR 2025
>
> [B] Luo, Yunhao and Yilun Du. “Grounding Video Models to Actions through Goal Conditioned Exploration.” ICLR 2025
>
> [C] Luo, Calvin et al. “Solving New Tasks by Adapting Internet Video Knowledge.” ICLR 2025
>
> [D] Tang, Weiliang et al. “Embodiment-Agnostic Action Planning via Object-Part Scene Flow.”

---

> ### Author Response · Authors · 2025-11-18
> **Is note on Ethics Concerns a mistake?**
>
> We believe the paragraph under "Details Of Ethics Concerns" is a mistake but would like to verify with the reviewer about this again.
>
>
> Also, on high level semantic understanding ability, we direct the reviewer to *Figure 6: Semantic Awareness Visualization* in appendix which contains several relevant examples.

---

> ### Comment · Reviewer_CeJz · 2025-11-18
>
> First, thank you for the authors’ response, and I apologize for mistakenly putting the weaknesses summary under Details of Ethics Concerns. Considering the authors’ reasonable explanations regarding the Calvin benchmark, I will raise my score. However, I do not fully agree with some of the authors’ points.
>
> UniPi is the first work to apply video generation models to the VLA domain. The fact that it was previously accepted by a conference does not mean that its high latency issues do not exist or can be ignored, nor was my decision to reject the paper based solely on that issue.
>
> Second, for VLA papers, I do not recommend using MetaWorld as the main simulation benchmark. Although MetaWorld includes tasks such as opening drawers that require instruction disambiguation, these tasks can often be distinguished simply by their initial states—this is true for most MetaWorld tasks—so little semantic understanding is actually required.
>
> Third, the real-world experimental setup appears relatively simple. I only saw demonstrations for the pick-and-place task. Can LangToMo perform other more complex tasks?

---

> ### Author Response · Authors · 2025-11-19
> **Continued Discussion**
>
> We thank the reviewer for the quick response, valuable feedback, and increased score. We are highly grateful for your time in joining this discussion. We further address the 3 questions below.
>
> &nbsp;
>
> **1. High Latency Issues**
>   - We understand high latency is a drawback of our default method. We will highlight it better during our revision.
>   - Our default setup uses a 25-step DDIM schedule for denoising during inference. We also explored a variant using a single denoising step (motivated by VPP [1]). System-2 training remains unchanged. We use additional joint fine-tuning of our System-1 (using System-2 outputs instead of GT optical flows). The intermediate feature is no longer visually as meaningful (the generated optical flow looks very noisy) and we lose the decoupled training advantage of our framework.
> However, this leads to almost *25x speedup in our runtime* with minimal change in performance. We will highlight this variant further in our main paper.
>   - We detail runtimes of our method, new variant, and related baselines below.
>
> | Method         | Denoising Steps | Runtime | CALVIN (ABC 10%) | MetaWorld |
> |----------------|:---------------:|:-------:|:----------------:|:---------:|
> | Ours (default) |        25       |   3.8s  |       3.38       |    57.7   |
> | Ours (variant) |        1        |  0.16s  |       3.36       |    57.1   |
> | VPP            |        1        |  0.21s  |       3.25       |     -     |
> | Pi0            |        -        |  0.59s  |         -        |    2.7    |
>
>   - We did not highlight this latency aspect in our paper earlier, since we considered this outside the focus of this work. However, after the feedback from reviewer, we believe this is important and are working on highlighting this in our revision.
>
> &nbsp;
>
> **2. Simulation Benchmark**
>   - After discussion among the authors, we have decided to move the CALVIN results into our main paper and highlight these as our main simulation results. Thank you again for this suggestion.
>
> &nbsp;
>
> **3. Real World Tasks**
>   - We believe LangToMo will generalize to complex tasks in the real world, since it is able to perform complex tasks in our simulation benchmarks (e.g. CALVIN, MetaWorld, iThor).
>   - Our current real world tasks are limited to pick-and-place tasks, but involve differing backgrounds with distractor objects and placement points to demonstrate semantic awareness.
>   - We are also able to perform pick-and-place beyond table-top in the 3D world as illustrated in our first page figure.
>   - Currently, we are not able to evaluate any baselines or LangToMo on more complex tasks in the real world due to limitations in our real world environment setup. We apologize for this.
>   - Also note that most robotics works published at recent conferences, including at ICLR '25, focus on similar real world tasks due to the lack of formalized real world benchmarks and difficulty of setup for complex tasks. In fact, all our real world experiments are modeled after an ICLR '25 paper (see L356-357), following their setup and protocol identically (with setup verification through correspondence with authors of that ICLR'25 paper).
>   - We will highlight in the revision that our real world tasks are limited to pick-and-place, and that we hope to explore more complex tasks in future work.
>
>
> &nbsp;
>
> [1] Hu, Yucheng, et al. "Video prediction policy: A generalist robot policy with predictive visual representations." ICML 2025

---

> > ### Author Response · Authors · 2025-11-25
> > **Updated PDF**
> >
> > Thank you again for the valuable feedback.
> >
> > We have uploaded our revised manuscript incorporating the suggested updates. We would be highly grateful if you are able to review these changes and take these into consideration in your final decision.
> >
> > We apologize again for the lack of clarity on our end that led to several misunderstandings in the beginning.

---

> > > ### Author Response · Authors · 2025-11-26
> > > **Remaining Barriers to Acceptance**
> > >
> > > Thank you again for the active engagement and the score increase. We are glad our clarifications on CALVIN were helpful. We have uploaded the revised PDF (changes in green) with all discussed modifications.
> > >
> > > We noticed the score is currently a 4. Given that we have addressed the primary concerns raised in the initial review:
> > >
> > >   - **Latency:** Addressed via the 25x speedup variant.
> > >   - **Benchmarks:** Highlighted CALVIN results in the main paper.
> > >   - **Real World:** Demonstrated the validity of our real-world experiments.
> > >
> > > Could you kindly clarify what major barrier remains preventing a positive recommendation? We are eager to address any final hesitation regarding task complexity or other aspects in this final week.

---

> > > > ### Comment · Reviewer_CeJz · 2025-11-28
> > > >
> > > > Thank you for your detailed supplementary experiments and revisions. It’s great to see that the model still delivers solid performance at a higher frequency. Based on the comprehensive results, I will raise the score to 6. However, the innovation of the paper is indeed relatively limited at present, so a higher score may not be feasible. It is hoped that more challenging experimental setups can be designed in the future to make up for this.

---

> ### Author Response · Authors · 2025-11-28
> **Thank you for the Continued Discussion & Reconsideration**
>
> We sincerely thank you for your active engagement throughout the review process and for making several valuable suggestions on latency, benchmarks, and evaluation. We are glad that our additional experiments on inference latency and the revised benchmarks helped address your concerns.
>
> We will further update the manuscript to highlight limitations in current real-world experiments, while noting that our setup aligns with several other prior works published at ICLR. We agree with the importance of more challenging experimental setups and are already exploring this for future work.
>
> Regarding innovation, we acknowledge the complex tasks handled by recent works like Pi0 (LLM backbone) and VPP (diffusion backbone), as well as our architectural similarities to diffusion backbone based works like AVDC and VPP. However, we respectfully suggest that our core contribution lies in demonstrating the efficacy of predicting pixel motion (as an alternative to RGB). We believe establishing its superiority, particularly in low-data regimes, offers a valuable and distinct perspective to the field.

---

### Official Review · Reviewer_AqBk · 2025-10-30

**Soundness:** 2
**Presentation:** 2
**Contribution:** 1
**Rating:** 2
**Confidence:** 3

**Summary:**

This paper proposes to use pixel motion as a control interface. System 1 translate language into pixel motions while system 2 translate the motions into robot actions. The authors show performance gain in MetaWorld experiments.

**Strengths:**

The writing is clear and easy to follow.

The method is able to leverage the unlabeled human data and enable scaled learning.

**Weaknesses:**

The novelty is limited. Many papers have explored the idea of extracting universal action representation from videos.

The performance is only evaluated on MetaWorld and the performance gain is marginal compared to ATM. More evaluations are needed.

See questions below.

**Questions:**

(1)	Another line of works that using latent actions to capture the pixel motions is missing, including works like LAPA, IGOR, Villa-X, UniVLA, etc. Comparison and discussion are needed.

(2)	As for the experiments, the author pretrain the models on the robot data (L361) and finetune the models on robot and human data. However, the pixel motions, the setting should be that we have many unlabeled video data (of human) and a small amount of labeled robot data.

(3)	In Table 5, why running two systems at the same frequency will lead to a performance drop?

---

> ### Author Response · Authors · 2025-11-18
> **Rebuttal to Reviewer AqBk**
>
> **1. Novelty**
>   - First to directly predict optical flow, the motion of every pixel in an image, conditioned on language without using any heuristic based point selection. No prior work performs this.
>   - First to construct a dual-system framework for robot control using such dense optical flow intermediate features generated conditioned on language goal and past motion information.
>   - First to improve such optical flow prediction from cheap-to-collect human demos of the downstream real world tasks.
>
> &nbsp;
>
> **2. Evaluations**
>   - ATM uses action labelled demos for fine-tuning. We outperform them marginally (+0.1) without using any action label demos. When using action labelled demos, we outperforms them by over +5.7% increase in success rate.
>   - We provide extensive evaluations on Real World environments in Tables 1 & 2.
>   - We provide experiments on CALVIN and iThor datasets in the appendix.
>   - As explained in L403-404, “We select this (MetaWorld) environment and tasks for direct comparison to Ko et al. (2023), which is the closest prior work to our method …”. This is the reason we highlight MetaWorld results in the main paper and move other results to appendix.
>
> &nbsp;
>
> **Q1. Latent action representations**
>   - Thank you for these valuable references. We will discuss these in our related work during revision.
>   - While methods like LAPA, IGOR, Villa-X, UniVLA allow pretraining from actionless videos, their latent representations require fine-tuning on robot action trajectories. This is in contrast to interpretable optical flow representation which allows action-free control in methods like ours and AVDC.
>   - We also investigate and highlight the ability to benefit from human demos (much cheaper to collect than robot demos) in downstream real world tasks, distinguishing ours from these prior works.
>   - Our optical flow intermediate features also allow decoupled training of each system. This is because optical flow is an interpretable, well-defined representation that can be calculated deterministically from frame pairs.
>
> &nbsp;
>
> **Q2. Ideal setting for robotics**
>   - “the setting should be that we have a small amount of labeled robot data” - we use zero labeled robot data for LTM-H (Table 2, 3, 4). We use minimal labelled data for LTM-S (Table 4).
>   - In contrast to prior work, our unlabelled human video data is collected in the real world downstream tasks. This setting is explored in several other contemporary works too, e.g. see Gr-3 technical report [A], because human demos are cheaper to collect than robot demos.
>   - In this work, our scope is to explore the usefulness of optical flow representations. We believe that we establish its usefulness through extensive real world experimentation (Table 2, 3) and comparison to multiple related prior work on simulated benchmarks (Table 4, 7, 8).
>   - Thank you for the suggestion of that setting (pretrain on more human videos) - it is definitely interesting to explore that, however as described in our paper, it is not a scope we explore for this paper.
>
> &nbsp;
>
> **Q3. In Table 5, why running two systems at the same frequency will lead to a performance drop?**
>   - Our hypothesis is that the optical flows predicted by Sys2 cannot be mapped to a single low level action.
>
> &nbsp;
>
> [A] Cheang, Chi-Lam et al. “GR-3 Technical Report.” ArXiv abs/2507.15493 (2025)

---

### Official Review · Reviewer_9Laj · 2025-10-31

**Soundness:** 2
**Presentation:** 3
**Contribution:** 2
**Rating:** 4
**Confidence:** 3

**Summary:**

This paper proposes LangToMo, a dual-system vision-language-action (VLA) framework that uses pixel motion (optical flow) as a universal intermediate representation for robot control. System 2 is a language-conditioned diffusion model that predicts pixel motion from a single image and instruction and system 1 is a  mapping function that converts the generated pixel motion into robot actions.

**Strengths:**

1. The paper identifies a key bottleneck in robot learning from videos: the need for action supervision and embodiment-specific data. The idea of treating pixel motion as a universal, interpretable, and embodiment-agnostic abstraction is good.
2. Dual-system design also satisfied the real-time issue of robot policy

**Weaknesses:**

1. Although the idea of using pixel motion as action representation is nice, I feel the idea is widely studied in previous work. The author list the difference with previous works at Table 1. I feel the idea is a little bit incremental.
2. For the simulation experiments, the author only did experiments on 11 Metaworld benchmarks tasks, which is limited. Many previous works train language conditioned policy on the whole Metaworld benchmark. Also, Metaworld is not designed for language-conditioned tasks, maybe run methods on Calvin or Libera can better verify the effectiveness of the method.

**Questions:**

1. Could you include comparisons with more advanced vision-language-action (VLA) models? Since the proposed approach ultimately produces a language-conditioned policy, it should also be evaluated against general VLA policies, not only motion-based ones?

---

> ### Author Response · Authors · 2025-11-18
> **Rebuttal to Reviewer 9Laj**
>
> **1. Novelty**
>   - First to directly predict optical flow, the motion of every pixel in an image, conditioned on language without using any heuristic based point selection. No prior work performs this.
>   - First to construct a dual-system framework for robot control using such dense optical flow intermediate features generated conditioned on language goal and past motion information.
>   - First to improve such optical flow prediction from cheap-to-collect human demos of the downstream real world tasks.
>
> &nbsp;
>
> **2a. MetaWorld Selection**
>   - As explained in L403-404, “We select this (MetaWorld) environment and tasks for direct comparison to Ko et al. (2023), which is the closest prior work to our method …”. This is the reason we highlight MetaWorld results in the main paper and move other results to appendix.
>   - These 11 MetaWorld tasks, proposed and used in Ko et al (NeurIPS 2023), are a standard split used across several works operating under “actionless” training settings [A,B,C,D].
>
> &nbsp;
>
> **2b. CALVIN results**
>   - As shown in our appendix (Table 7), we achieve strong results on the CALVIN 10% ABC->D setting outperforming all prior work, using a much smaller model and significantly less pretraining than some top methods.
>   - We also outperform all but two methods (VPP, DreamVLA)  in the CALVIN 100% ABC->D setting.
>   - A scaled up variant of our model with pretraining matching VPP outperforms both VPP and DreamVLA, achieving 4.48 average length on CALVIN.
>   - However, we reiterate that the purpose of this paper was to establish the usefulness of our framework for real world robot manipulation tasks, demonstrated by Table 2 & 3.
>   - We also consider our Metaworld results in Table 4 (on a standard benchmark used in actionless settings) as sufficient comparison against prior work exploring similar pixel-trajectory modelling for robotic tasks.
>
>
> &nbsp;
>
> **Q1. Comparisons with more advanced VLA models?**
>   - All our real world experiments use the advanced VLA from LLaRA (ICLR '25) as the primary baseline. This is explained in Sec 4.1 and Tables 2 & 3.
>   - VLAs like Pi0 are much slower and cannot operate under our actionless setting.
>   - On our limited action trajectory data setting (i.e. <20 demos per task), VLAs perform extremely poorly, achieving close to random results.
>   - We provide results for Pi0 (finetune only diffusion head) vs motion based methods on MetaWorld below.
>
> |   Method | door-open | door-close | basketball | shelf-place | btn-press | btn-top | faucet-close | faucet-open | handle-press | hammer | assembly | Overall |
> |------------------------------|:---------:|:----------:|:----------:|:-----------:|:---------:|:-------:|:------------:|:-----------:|:------------:|:------:|:--------:|:-------:|
> | Im2Flow2Act                  |    0.0    |     0.0    |     0.0    |     4.0     |    6.3    |   0.0   |      7.3     |     4.7     |      0.0     |   0.0  |    0.0   |   2.0   |
> | ATM                          |    75.3   |    90.7    |    24.0    |     16.3    |    77.3   |   76.7  |     50.0     |     62.7    |     92.3     |   4.3  |    2.0   |   52.0  |
> | AVDC               |    72.0   |    89.3    |    37.3    |     18.7    |    60.0   |   24.0  |     53.3     |     24.0    |     81.3     |   8.0  |    6.7   |   43.1  |
> | Pi0                  |    0.0    |     0.0    |     0.0    |     5.0     |    8.4    |   0.0   |      6.8     |     9.2     |      0.0     |   0.0  |    0.0   |   2.7   |
> | LTM-S (Ours) |    77.3   |    95.0    |    39.0    |     20.3    |    82.7   |   84.3  |     52.3     |     68.3    |     98.0     |  10.3  |    7.7   |   57.7  |
>
>
>
> &nbsp;
>
> **Q2. VLA Comparison**
>   - All our real world experiments use the VLA method LLaRA (ICLR '25) as the primary baseline.
>   - Our language conditioned policy is trained without action trajectory data
>   - Our method can also be deployed without training on action trajectory data leveraging the interpretable nature of optical flow (similar to prior work, e.g. AVDC)
>   - Some VLA models (e.g. Pi0) do not operate under this settings - they need thousands of robot trajectory training data
>
> &nbsp;
>
> [A] Luo, Calvin et al. “Solving New Tasks by Adapting Internet Video Knowledge.” ICLR 2025
>
>
> [B] Luo, Yunhao and Yilun Du. “Grounding Video Models to Actions through Goal Conditioned Exploration.” ICLR 2025
>
>
> [C] Luo, Calvin et al. “Solving New Tasks by Adapting Internet Video Knowledge.” ICLR 2025
>
>
> [D] Tang, Weiliang et al. “Embodiment-Agnostic Action Planning via Object-Part Scene Flow.”

---

> ### Author Response · Authors · 2025-11-26
> **Follow-up: VLA Comparisons (Pi0) and CALVIN Results**
>
> As the discussion period closes soon, we wanted to ensure you had a chance to review our new results addressing your two main requests: comparisons to advanced VLAs and more benchmarks (CALVIN).
>
> &nbsp;
>
> 1. **Comparison to Advanced VLAs (Pi0 & LLaRA)**
>     - **Simulation (Pi0):** We ran the comparison against Pi0 (a state-of-the-art VLA) on MetaWorld. As shown in the table in our previous comment, Pi0 fails completely (2.7% success rate) in this low-data fine-tuning setting, while our method achieves 57.7%. This highlights that our method solves a distinct problem (learning from video without heavy action supervision) that standard VLAs cannot handle.
>     - **Real World (LLaRA):** We also highlight our real-world comparisons against the advanced VLA method LLaRA (ICLR '25), which we consistently outperform (Tables 2 & 3 in the paper).
>
> &nbsp;
>
> 2. **Benchmarks (CALVIN):** Regarding your suggestion to use CALVIN, we have now moved these results to the main paper. We repeat them below for your quick reference. Strength of our approach is again more visible in the low data regime (10% training data) for this benchmark as well.
>
>
> &nbsp;
>
> | Method           | Training Data | Task 1 | Task 2 | Task 3 | Task 4 | Task 5 | Avg. Len |
> |------------------|:-------------:|:------:|:------:|:------:|:------:|:------:|:--------:|
> | GR-1             |    10\% ABC   |  0.672 |  0.371 |  0.198 |  0.108 |  0.069 |   1.41   |
> | VPP              |    10\% ABC   |  0.878 |  0.746 |  0.632 |  0.540 |  0.453 |   3.25   |
> | LTM-S (ours)     |    10\% ABC   |  0.896 |  0.769 |  0.652 |  0.596 |  0.467 |   **3.38**   |
>
> &nbsp;
>
> | Method           | Training Data | Task 1 | Task 2 | Task 3 | Task 4 | Task 5 | Avg. Len |
> |------------------|:-------------:|:------:|:------:|:------:|:------:|:------:|:--------:|
> | RT-1             |   100\% ABC   |  0.533 |  0.222 |  0.094 |  0.038 |  0.013 |   0.90   |
> | Diffusion Policy |   100\% ABC   |  0.402 |  0.123 |  0.026 |  0.008 |  0.00  |   0.56   |
> | Robo-Flamingo    |   100\% ABC   |  0.824 |  0.619 |  0.466 |  0.331 |  0.235 |   2.47   |
> | Uni-Pi           |   100\% ABC   |  0.560 |  0.160 |  0.080 |  0.080 |  0.040 |   0.92   |
> | MDT              |   100\% ABC   |  0.631 |  0.429 |  0.247 |  0.151 |  0.091 |   1.55   |
> | Susie            |   100\% ABC   |  0.870 |  0.690 |  0.490 |  0.380 |  0.260 |   2.69   |
> | GR-1             |   100\% ABC   |  0.854 |  0.712 |  0.596 |  0.497 |  0.401 |   3.06   |
> | Vidman           |   100\% ABC   |  0.915 |  0.764 |  0.682 |  0.592 |  0.467 |   3.42   |
> | RoboUniview      |   100\% ABC   |  0.942 |  0.842 |  0.734 |  0.622 |  0.507 |   3.65   |
> | LTM-S (ours)     |   100\% ABC   |  0.971 |  0.824 |  0.728 |  0.672 |  0.606 |   **3.81**   |
>
> &nbsp;
>
> We believe these results directly address your concerns regarding baselines and benchmarks. Could you let us know if these clarifications resolve your hesitation regarding our contribution?

---

### Author Response · Authors · 2025-11-18
**Two key clarifications to resolve misunderstanding**

We thank reviewers for all positive feedback.
  - *Addresses an important area and offers insightful directions for future research.*
  - *The paper is well written, the method is clearly presented, and the figures/tables are complete and easy to read.*
  - *Dual-system design also satisfied the real-time issue of robot policy*
  - *The writing is clear and easy to follow.*
  - *The method is able to leverage the unlabeled human data and enable scaled learning.*
  - *Surpass other baseline methods in real world zero-shot tasks via large-scale pretraining.*


&nbsp;

We are surprised to see the reject ratings given the highly positive feedback. We believe there have been some misunderstandings regarding our work, and focus on the following two points to allow better clarification. We apologize in advance for any lack of clarity on our part that led to this misunderstanding.

&nbsp;

**1. Pixel Motion = Future Optical Flow**
  - From a single frame, we predict future optical flow (motion for every pixel in image). We realize our choice of wording “pixel motion” is possibly unclear. We will highlight this clearly during revision.
  - No prior work explores directly learning this from videos conditioned on text. They use subsets of image pixels, picked using task-specific heuristics, which makes training less scalable.

&nbsp;

**2. No robot action data for training**
  - Our real world results (Table 2 & 3) as well as MetaWorld results (Table 4 LTM-H) are not trained on any action trajectory data.
  - We specifically pick this MetaWorld split because most prior works operating under this “actionless” training setting (e.g. AVDC) use this split.
  - VLA models like OpenVLA & Pi0 cannot operate under this setting. They need thousands of extensive robot action trajectory data to pretrain followed by hundreds of downstream task robot demos for finetuning.
  - Even in our downstream finetuning setting (Table 4 LTM-S), we use as few as 10-20 robot demos for finetuning.

---

### Author Response · Authors · 2025-11-24
**Updated Manuscript**

A revised version of our manuscript has now been uploaded (all revisions in green).


We thank Reviewer CeJz for their engagement that resulted in several of the modifications.
We would be highly grateful if all reviewers could further engage in the discussion.

&nbsp;

We believe that several reviewers have misunderstood key ideas in the paper:
  - pixel motion definition
  - our actionless training setting

We hope our rebuttal clarifies these and that discussion could further resolve any misunderstandings.

---

### Author Response · Authors · 2025-12-02
**Rebuttal Summary for New Area Chair**

1. **Confirmed Resolution (Scores Updated):**
After our clarifications on (a) pixel motion = future optical flow over all pixels and (b) our actionless training setting and benchmark choices, *two reviewers explicitly raised their scores from 2 $\rightarrow$ 6*:

    - **YQmb** (*"increase my rating to 6"*): concerns on motivation/novelty and simulation strength were addressed by the above clarifications as well as our comparisons to prior dual-system works and visual-trace/flow-subset baselines.
    - **CeJz** (*"raise the score to 6"*): concerns on latency, MetaWorld as main benchmark, and CALVIN/real-world strength were addressed via highlighting the efficient 1-step variant, moving CALVIN into the main paper, and expanded VLA comparisons.

&nbsp;

2. **Pending but Aligned Concerns (AqBk, 9Laj):**
The remaining reviewers’ objections mirror those already resolved for YQmb/CeJz, namely novelty of pixel motion vs prior video/latent-action work and benchmark breadth/VLA baselines. Our rebuttal and revised paper provide the same clarifications (dense language-conditioned optical flow, actionless setting, CALVIN + VPP/Pi0/LLaRA results).

&nbsp;

3. **For the AC’s Consideration:**
The core “reject” arguments (novelty and benchmarks) were addressed during discussion, which resulted in two reviewers updating their scores from 2 $\rightarrow$ 6. We respectfully highlight this evolution for the AC’s consideration.

---

### Note · Authors · 2026-02-01

**Comment:**

This post is for the purpose of formally noting several *factual oversights* in the Area Chair’s meta-review.
We believe this would allow a more fair and transparent record of our paper's evolution, despite the unfortunate breakdown and early halting of the review process.

&nbsp;

### Mischaracterization of Final Reviewer Consensus
The meta-review states that “three out of four reviewers remained unconvinced” and that concerns regarding novelty and generalizability were not sufficiently addressed. This characterization is factually incorrect based on the final state of the discussion:
  - Two reviewers (CeJz, YQmb) explicitly raised their scores to 6 after detailed discussion and additional experiments.
  - Reviewer 9Laj indicated they “would not mind if the paper is accepted” (rating 4).
  - Only one reviewer (AqBk) maintained a firm reject.

Thus, the final reviewer consensus was not overwhelmingly negative, and in fact showed clear positive movement after rebuttal and revisions, despite the early termination of the review period, which prevented further discussion.

&nbsp;

###  Resolution of Key Reviewer Concerns Not Acknowledged
The meta-review repeats concerns about benchmark breadth and real-world task simplicity without acknowledging that:
  - Real-world experiments strictly followed protocols from prior ICLR papers, a point explicitly clarified during discussion.
  - CALVIN results were moved into the main paper per reviewer request and showed competitive or state-of-the-art performance, particularly in low-data regimes.
  - A 25× faster inference variant was added and documented, directly addressing latency concerns.

These points were discussed at length in the review thread and explicitly acknowledged by multiple reviewers, yet they are not reflected in the meta-review.

&nbsp;

### Novelty Evaluation Misaligned with Final Reviews

The claim that the paper’s technical novelty is “limited” is presented without engaging with the paper’s central contribution—namely, the use of dense, language-conditioned optical flow over all pixels as an interpretable, embodiment-agnostic intermediate representation trained in an actionless setting. This distinction was clarified repeatedly and ultimately accepted by multiple reviewers, but appears not to have been reflected in the final recommendation.

&nbsp;

### Thank you

We understand the difficultly of the AC process, especially in this cycle given the unfortunate incidents during reviewing. We respect the difficulty of the AC role and are thankful to the AC for their service. We value all time and resources devoted towards evaluating our work.
Nonetheless, given the documented reviewer score increases and resolved concerns, we believe the final decision and comments do not accurately reflect the substance of the discussion or the final state of reviewer consensus.
This note is simply for the purpose of documenting our differing perspective.

We are therefore withdrawing the paper and will pursue submission at a venue where the clarified contributions and experimental evidence can be evaluated afresh.

**Withdrawal Confirmation:**

I have read and agree with the venue's withdrawal policy on behalf of myself and my co-authors.

---

### Meta-Review · Area_Chair_WNLE · 2026-01-10

**Summary:**

The primary concerns focus on limited technical novelty and insufficient evidence of generalizability beyond simple tasks.
- Three out of four reviewers remained unconvinced that the proposed method offers a significant enough paradigm shift over existing ones.
- The distinction between the proposed method and prior work is marginal.
- The real-world evaluation was critiqued for being restricted to basic pick-and-place tasks, failing to demonstrate the universal power of the representation in more complex, long-horizon, or highly dynamic robotic scenarios.

**Reviewer Concerns:**

Addressed:
- Concern regarding diffusion speed has been addressed
- More baselines are included

Outstanding Concerns:
- Still, some reviewers believe the novelty of the paper is weak
- The universal claim is not sufficiently supported by the limited diversity of the physical experiments.

**Reviewer Scores:**

Two reviewers tend to increase their score. While the other are likely to keep the original ones.

---

### Decision · Program_Chairs · 2026-01-26

Reject